# Matrin 3-dependent neurotoxicity is modified by nucleic acid binding and nucleocytoplasmic localization

Ahmed M Malik[1,2], Roberto A Miguez[3], Xingli Li[3], Ye-Shih Ho[4], Eva L Feldman[2,3,5], Sami J Barmada[2,3]*

[1]Medical Scientist Training Program, University of Michigan, Ann Arbor, United States; [2]Neuroscience Graduate Program, University of Michigan, Ann Arbor, United States; [3]Department of Neurology, University of Michigan, Ann Arbor, United States; [4]Institute of Environmental Health Sciences, Wayne State University, Detroit, United States; [5]Program for Neurology Research and Discovery, University of Michigan, Ann Arbor, United States

**Abstract** Abnormalities in nucleic acid processing are associated with the development of amyotrophic lateral sclerosis (ALS) and frontotemporal dementia (FTD). Mutations in *Matrin 3* (*MATR3*), a poorly understood DNA- and RNA-binding protein, cause familial ALS/FTD, and MATR3 pathology is a feature of sporadic disease, suggesting that MATR3 dysfunction is integrally linked to ALS pathogenesis. Using a rat primary neuron model to assess MATR3-mediated toxicity, we noted that neurons were bidirectionally vulnerable to MATR3 levels, with pathogenic MATR3 mutants displaying enhanced toxicity. MATR3's zinc finger domains partially modulated toxicity, but elimination of its RNA recognition motifs had no effect on survival, instead facilitating its self-assembly into liquid-like droplets. In contrast to other RNA-binding proteins associated with ALS, cytoplasmic MATR3 redistribution mitigated neurodegeneration, suggesting that nuclear MATR3 mediates toxicity. Our findings offer a foundation for understanding MATR3-related neurodegeneration and how nucleic acid binding functions, localization, and pathogenic mutations drive sporadic and familial disease.
DOI: https://doi.org/10.7554/eLife.35977.001

*For correspondence:
sbarmada@umich.edu

Competing interests: The authors declare that no competing interests exist.

## Introduction

Amyotrophic lateral sclerosis (ALS) is a progressive neurodegenerative disorder resulting in the death of upper and lower motor neurons (*Charcot and Joffroy, 1869*). Mounting evidence indicates that RNA-binding proteins (RBPs) are integrally involved in the pathogenesis of ALS (*Taylor et al., 2016*). The majority (>95%) of ALS patients display cytoplasmic mislocalization and deposition of the RBP TDP-43 (TAR DNA/RNA-binding protein of 43 kDa) in affected tissue (*Neumann et al., 2006*). Moreover, over 40 different ALS-associated mutations have been identified in the gene encoding TDP-43, and mutations in several different RBPs have been similarly linked to familial ALS (*Kabashi et al., 2008*; *Kwiatkowski et al., 2009*; *Vance et al., 2009*; *Barmada and Finkbeiner, 2010a*; *Ticozzi et al., 2011*; *Kim et al., 2013*). These mutations often cluster in intrinsically disordered domains that facilitate reversible liquid-liquid phase separation (LLPS), thereby creating ribonucleoprotein granules important for RNA processing, shuttling of mRNAs to sites of local translation, or sequestration of transcripts during stress. Pathogenic mutations in the genes encoding TDP-43 and related RBPs, including FUS and TIA1, shift the equilibrium towards irreversible phase separation and the formation of cytoplasmic aggregates analogous to those observed in post-mortem tissues from patients with ALS (*Johnson et al., 2009*; *Patel et al., 2015*; *Gopal et al.,*

2017; *Mackenzie et al., 2017*). The downstream implications of abnormal LLPS on RNA misprocessing, RBP pathology, and neurodegeneration in ALS are unknown, however.

Matrin 3 (MATR3) is a DNA- and RNA-binding protein with wide-ranging functions in nucleic acid metabolism including gene transcription, the DNA damage response, splicing, RNA degradation, and the sequestration of hyperedited RNAs (*Belgrader et al., 1991*; *Hibino et al., 2000*; *Zhang and Carmichael, 2001*; *Salton et al., 2010*; *Coelho et al., 2015*; *Rajgor et al., 2016*; *Uemura et al., 2017*). The S85C mutation in *MATR3* leads to autosomal dominant distal myopathy with vocal cord and pharyngeal weakness (*Feit et al., 1998*; *Senderek et al., 2009*). A more recent report reclassified a subset of patients with this diagnosis as having ALS and noted several additional *MATR3* mutations in individuals with ALS and frontotemporal dementia (FTD), placing *MATR3* in a family of genes implicated in familial ALS, FTD, and myopathy. This family also includes *TIA1*, *VCP*, *p62/SQSTM1*, *hnRNPA1*, and *hnRNPA2/B1*, mutations in which lead to multisystem proteinopathy characterized by variable involvement of muscle and bone in addition to the central nervous system (*Hocking et al., 2002*; *Fecto et al., 2011*; *Kimonis et al., 2008*; *Johnson et al., 2010*; *Kim et al., 2013*; *Klar et al., 2013*; *Johnson et al., 2014*; *Mackenzie et al., 2017*). A total of 13 pathogenic *MATR3* mutations have now been identified, most of which result in amino acid substitutions within disordered stretches of the MATR3 protein (*Figure 1A*) (*Millecamps et al., 2014*; *Lin et al., 2015a*; *Origone et al., 2015*; *Leblond et al., 2016*; *Xu et al., 2016*; *Marangi et al., 2017*). Additionally, post-mortem analyses demonstrated MATR3 pathology—consisting of cytoplasmic MATR3 accumulation as well as strong nuclear immunostaining—in patients with sporadic ALS and familial disease due to *C9orf72* hexanucleotide expansions and *FUS* mutations (*Dreser et al., 2017*; *Tada et al., 2018*). Together, these observations suggest that MATR3 may be a common mediator of disease even in those without *MATR3* mutations.

Even so, little is known about MATR3's functions in health or in disease, and the mechanisms underlying MATR3-dependent neurotoxicity remain unclear. Here, we establish an in vitro model of MATR3-mediated neurodegeneration and take advantage of this model to investigate the intrinsic properties and domains of MATR3 required for toxicity. Furthermore, we examine how disease-associated MATR3 mutations affect these properties to enhance neurodegeneration.

## Results

### MATR3 levels modulate neuronal survival in an in vitro model of neurodegeneration

We first asked how MATR3 expression is related to neurodegeneration using longitudinal fluorescence microscopy (LFM), a sensitive high-content imaging system that we assembled for assessing neuronal function and survival at the single-cell level. As *MATR3* mutations cause a spectrum of disease that includes ALS and FTD, we modeled neurotoxicity in primary mixed cortical cultures, a system that recapitulates key features of ALS/FTD pathogenesis (*Barmada et al., 2010b*, *2014*, *2015*). Primary neurons were transfected with diffusely localized mApple to enable visualization of neuronal cell bodies and processes by fluorescence microscopy. In addition, cells were co-transfected with constructs encoding enhanced green fluorescent protein (EGFP) or MATR3 fused with EGFP. Cultures were imaged by fluorescence microscopy at 24 hr intervals for 10 days, and custom scripts (https://github.com/barmadaslab/survival-analysis and https://github.com/barmadaslab/measurements; *Miguez, 2018a*; *Miguez, 2018b* copies archived at https://github.com/elifesciences-publications/survival-analysis and https://github.com/elifesciences-publications/measurements) used to generate uniquely labeled regions of interest (ROIs) corresponding to each cell (*Figure 1B*). Rounding of the soma, retraction of neurites or loss of fluorescence indicated cell death; these criteria proved to be sensitive markers of neurodegeneration in previous studies (*Arrasate and Finkbeiner, 2005*). We used the time of death for individual cells to calculate an overall risk of death, expressed as a hazard ratio (HR), corresponding to the likelihood of cell death in each population relative to a control or reference group (*Christensen, 1987*). In doing so, we observed that MATR3-EGFP overexpression significantly increases the risk of death compared to EGFP alone, with a HR of 1.48 (*Figure 1C*).

Next, we investigated the dose-dependency of this MATR3 toxicity. Transient transfection delivers a different amount of vector to each cell, resulting in substantial variability in protein expression

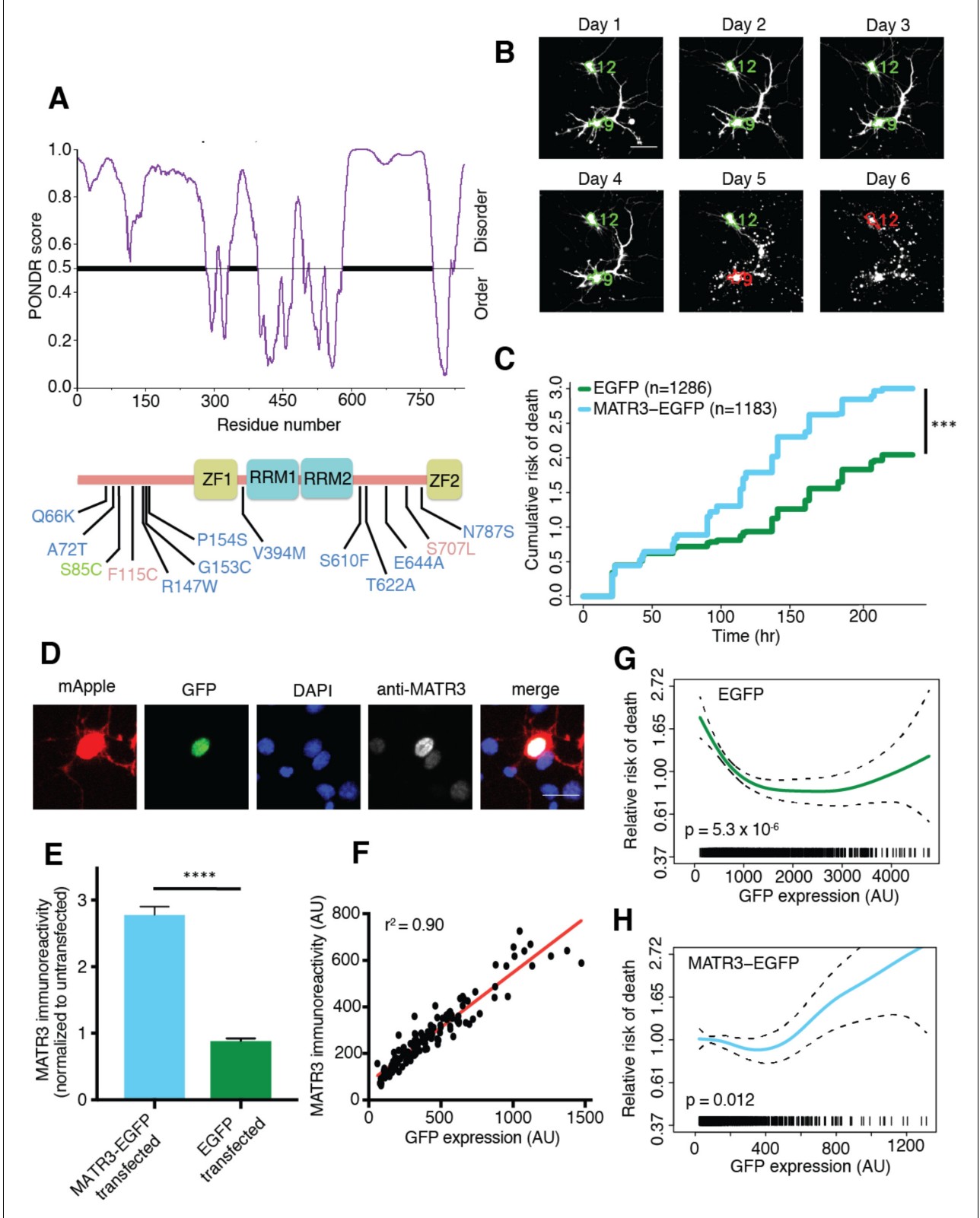

**Figure 1.** MATR3 overexpression results in dose-dependent neurodegeneration. (**A**) Diagram of MATR3 showing nucleic acid-binding domains as well as the distribution of pathogenic mutations implicated in ALS (blue), ALS/FTD (red), and ALS/distal myopathy (green) within domains predicted to be disordered by PONDR VSL2 (*Peng et al., 2006*). (**B**) Longitudinal fluorescence microscopy (LFM) allows unique identification and tracking of thousands of primary neurons (green outlines) transfected with fluorescent proteins, as well as monitoring of cell death (red outlines), indicated by loss of

*Figure 1 continued on next page*

*Figure 1 continued*

fluorescence signal and changes in morphology. (C) MATR3-EGFP expressing neurons exhibited a higher risk of death compared to neurons expressing only EGFP, as quantified by the hazard ratio (HR) (HR = 1.48; EGFP n = 1286, MATR3-EGFP n = 1183; ***$p<2\times10^{-16}$, Cox proportional hazards). (D–E) Transfection of neurons with MATR3-EGFP resulted in a 2.8-fold increase in anti-MATR3 immunoreactivity over untransfected cells (MATR3-EGFP n = 133, untransfected n = 136, EGFP n = 113, ****$p<0.0001$, one-way ANOVA with Tukey's post-hoc test). (F) On a single-cell basis, GFP fluorescence is directly proportional to anti-MATR3 reactivity ($p<0.0001$, $r^2 = 0.90$; linear regression). (G) Penalized spline modeling confirmed a protective effect associated with higher EGFP expression that plateaus at ~1500 arbitrary units (AU); ($p=5.3\times10^{-6}$, penalized spline regression). (H) However, penalized spline analysis showed no relationship between expression and survival at low and medium expression but a significant increase in risk of death with high MATR3-EGFP levels ($p=0.012$; penalized spline regression). Scale bars in (B) and (D), 20 μm.
DOI: https://doi.org/10.7554/eLife.35977.002

The following source data is available for figure 1:

**Source data 1.** Survival and GFP intensity data for *Figures 1C,G, and H* and GFP intensity and immunoreactivity data used for *Figures 1E and F*.
DOI: https://doi.org/10.7554/eLife.35977.003

for individual cells. Such variability can be difficult to appreciate using population-based approaches such as Western blotting but are readily visualized by single-cell techniques including immunofluorescence (*Arrasate et al., 2004*; *Miller et al., 2010*; *Barmada et al., 2015*). Therefore, to estimate the degree of MATR3 overexpression in individual neurons, we measured MATR3 antibody reactivity by quantitative immunofluorescence in neurons transfected with EGFP or MATR3-EGFP (*Figure 1D*). There was no significant difference in MATR3 antibody reactivity between EGFP transfected and untransfected cells (*Figure 1E*). In comparison, MATR3-EGFP transfected cells showed a 2.8-fold increase in MATR3 antibody reactivity compared to untransfected cells. Further, and in agreement with previous work relating single-cell fluorescence intensity to immunoreactivity (*Arrasate et al., 2004*), we detected a linear relationship between EGFP fluorescence intensity and anti-MATR3 antibody reactivity in individual neurons expressing MATR-EGFP (*Figure 1F*). These data confirm that GFP intensity provides a reliable, single-cell estimate of EGFP or MATR3-EGFP expression.

We took advantage of this relationship to analyze the association between EGFP or MATR3-EGFP expression (measured 24 hr after transfection) and neuronal survival using penalized splines (*Miller et al., 2010*; *Barmada et al., 2015*). These models enable us to predict the impact of single-cell protein expression on the risk of death within separate populations of cells expressing either EGFP (*Figure 1G*) or MATR3-EGFP (*Figure 1H*). Consistent with the results of prior studies, we detected a reduced risk of death in association with higher EGFP expression levels (*Miller et al., 2010*), implying that unhealthy or dying neurons are unable to express high amounts of EGFP. Conversely, we noted a significant increase in the risk of death for cells expressing high levels of MATR3-EGFP (*Figure 1H*); this relationship is similar to that observed for other proteins associated with neurodegenerative disorders, including TDP-43 (ALS/FTD; *Barmada et al., 2015*) and mutant huntingtin (Huntington's disease; *Miller et al., 2010*). Taken together, these data support a dose-dependent toxicity of MATR3-EGFP in primary neurons.

Several *MATR3* mutations are responsible for familial ALS, FTD, and hereditary distal myopathy (*Senderek et al., 2009*; *Johnson et al., 2014*; *Millecamps et al., 2014*; *Lin et al., 2015a*; *Origone et al., 2015*; *Leblond et al., 2016*; *Xu et al., 2016*; *Marangi et al., 2017*). To determine if disease-associated *MATR3* mutations accentuate neurodegeneration, we created MATR3-EGFP fusion proteins harboring one of four mutations originally implicated in familial disease: S85C, F115C, P154S, and T622A (*Figure 1A*). Primary rodent cortical neurons expressing these mutant MATR3-EGFP constructs exhibited the same granular nuclear distribution as MATR3(WT)-EGFP, without obvious aggregation or cytoplasmic mislocalization, in accordance with prior reports (*Figure 2A*) (*Gallego-Iradi et al., 2015*; *Boehringer et al., 2017*). Even so, all four displayed a modest but significant increase in toxicity over MATR3(WT)-EGFP when overexpressed in primary neurons (*Figure 2B*), consistent with either gain-of-function or dominant negative loss-of-function mechanisms contributing to mutant MATR3-associated neurodegeneration.

To determine if loss of endogenous MATR3 function is sufficient for neurodegeneration, we transfected primary neurons with mApple and siRNA targeting the amino (N)-terminal coding region of rodent *Matr3* or a scrambled siRNA control. Three days after transfection, Matr3 immunoreactivity was used to quantify efficacy of knockdown in transfected cells (*Figure 2C*). Compared to scrambled siRNA-transfected cells, we noted consistent depletion of the endogenous rat Matr3 by

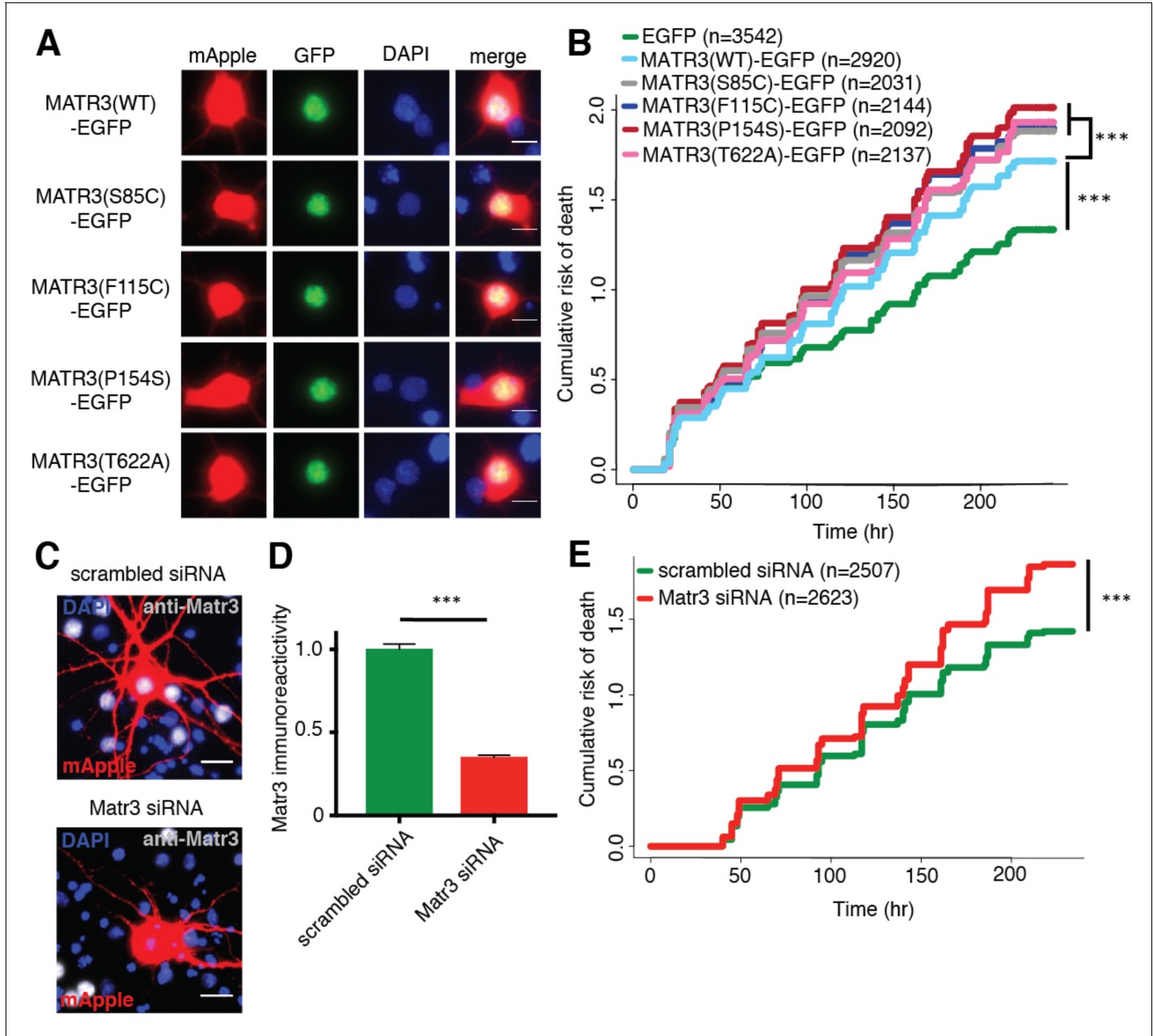

**Figure 2.** Neurons are susceptible to both gain-of-function and loss-of-function MATR3 toxicity. (**A**) In primary rodent cortical neurons, the S85C, F115C, P154S, and T622A disease-associated MATR3 mutants have the same granular nuclear distribution as MATR3(WT)-EGFP. (**B**) All four disease mutations display a subtle but significant increase in toxicity compared to MATR3(WT)-EGFP (comparing to MATR3(WT)-EGFP, n = 2920: MATR3(S85C)-EGFP, HR = 1.16, n = 2031, ***p=3.79×10$^{-6}$; MATR3(F115C)-EGFP, HR = 1.14, n = 2144, ***p=5.57×10$^{-5}$; MATR3(P154S)-EGFP, HR = 1.24, n = 2092, ***p=1.77×10$^{-11}$; MATR3(T622A)-EGFP, HR = 1.14, n = 2137, ***p=6.02×10$^{-5}$; Cox proportional hazards). (**C–D**) siRNA targeting the endogenous rat *Matr3* reduced MATR3 antibody reactivity by approximately 65% (scrambled siRNA, n = 576; anti-Matr3 siRNA, n = 508; p<0.0001, two-tailed t-test). (**E**) Neurons transfected with anti-Matr3 siRNA displayed a higher risk of death compared to those transfected with scrambled siRNA (HR = 1.20; scrambled siRNA, n = 2507; anti-Matr3, n = 2623; ***p=2.05×10$^{-8}$, Cox proportional hazards). Scale bars in (**A**), 10 μm; scale bars in (**C**), 20 μm.
DOI: https://doi.org/10.7554/eLife.35977.004

The following source data is available for figure 2:

**Source data 1.** Survival data for *Figure 2B and E* and immunoreactivity data for *Figure 2D*.
DOI: https://doi.org/10.7554/eLife.35977.005

approximately 65% in those transfected with siRNA targeting *Matr3* (*Figure 2D*). Having confirmed knockdown, we imaged a separate set of transfected cells for 10 days to assess the effect of *Matr3* knockdown on neuronal survival. In doing so, we observed a 20% increase in the risk of death upon *Matr3* depletion in comparison to scrambled siRNA (*Figure 2E*). These data suggest that neurons are vulnerable to both increases and decreases in MATR3 levels and function; further, pathogenic *MATR3* mutations may elicit neurodegeneration via gain- or loss-of-function mechanisms, or through elements of both.

## MATR3's zinc finger domains modulate overexpression toxicity, but its RNA recognition motifs mediate self-association

To identify the functional domains involved in MATR3-mediated neurodegeneration, we systematically deleted each of the annotated MATR3 domains and evaluated subsequent toxicity upon overexpression in primary neurons (*Figure 3A*). MATR3 has two zinc-finger (ZF) domains of the C2H2 variety, which bind DNA but may also recognize RNA and/or mediate protein-protein interactions (*Brayer et al., 2008*; *Burdach et al., 2012*). Deletions of ZF1, ZF2, or both had no observable effect on MATR3-EGFP localization (*Figure 3B*), and ZF1 deletion by itself did not significantly alter toxicity compared to full-length MATR3-EGFP. In contrast, ZF2 deletion, either in isolation or combined with ZF1 deletion, partially rescued MATR3-EGFP overexpression toxicity (*Figure 3C*).

We next created deletion variants of MATR3's RNA recognition motifs (RRMs) to test their contribution to MATR3-mediated neurodegeneration. As with the MATR3 ZF domains, RRMs are capable of recognizing both RNA and DNA (*Inagaki et al., 1996*). While deletion of RRM1 failed to affect MATR3-EGFP localization, we noted a striking redistribution of MATR3(ΔRRM2)-EGFP into intranuclear granules in a subset of transfected neurons (*Figure 3D*). Deletion of RRM1 in combination with RRM2 produced the same phenotype, suggesting that RRM2 normally prevents such redistribution. These nuclear granules formed by MATR3(ΔRRM2)-EGFP and MATR3(ΔRRM1/2)-EGFP were uniformly spherical in shape, and their presence was accompanied by a reduction in the intensity of diffusely-distributed MATR3 within the nucleus, suggesting that they represent hyperconcentrated MATR3 puncta. Evidence from previous studies indicates that RNA recognition by MATR3 may be largely—but not solely—driven by RRM2 (*Hibino et al., 2006*; *Salton et al., 2011*). Consistent with this, our finding that RRM2 deletion induces the formation of nuclear condensates suggests that RNA binding normally keeps MATR3 diffuse by preventing an intrinsic tendency for self-association. We detected no colocalization of MATR3(ΔRRM2)-EGFP or MATR3(ΔRRM1/2)-EGFP with markers of nucleoli, nuclear speckles, or PML bodies (*Figure 3—figure supplement 1*), indicating that MATR3 lacking its RRM2 does not join these organelles. Despite the dramatic shift in MATR3-EGFP distribution with RRM2 deletion, there was no associated change in the toxicity of MATR3-EGFP lacking RRM1, RRM2, or both in comparison to MATR3(WT)-EGFP (*Figure 3E*). This finding stands in contrast to what has been observed for other ALS/FTD-associated RBPs, for which the ability to bind RNAs is a key mediator of overexpression toxicity (*Elden et al., 2010*; *Daigle et al., 2013*; *Ihara et al., 2013*).

## The toxicity of RNA binding-deficient MATR3 variants is highly dependent on their subcellular distribution

One of the hallmarks of neurodegenerative diseases, including ALS and FTD, is the formation of protein-rich aggregates (*Arai et al., 2006*; *Neumann et al., 2006*). Prior investigations suggest that these aggregates may be toxic, innocuous, or representative of a coping response that ultimately prolongs neuronal survival (*Arrasate et al., 2004*; *Barmada et al., 2010b*). To determine if the formation of nuclear puncta by MATR3(ΔRRM2)-EGFP and MATR3(ΔRRM1/2)-EGFP affected neuronal lifespan, we turned to LFM. We employed a modified version of the automated analysis script to draw ROIs around the nuclear perimeter within each transfected cell (*Figure 4A*) and then calculated a coefficient of variation (CV) for the MATR3(ΔRRM1/2)-EGFP signal within each nuclear ROI. The CV, or the ratio of the standard deviation of GFP intensity to the mean GFP intensity for the ROI, is directly proportional to the spatial variability of fluorescence intensity within each ROI. Therefore, we reasoned that this measure might be useful for rapidly and reliably identifying puncta in an unbiased and high-throughput manner. We first validated the use of CV for detecting puncta by creating a receiver-operator characteristic (ROC) curve; in doing so, we observed that a CV threshold of 0.92

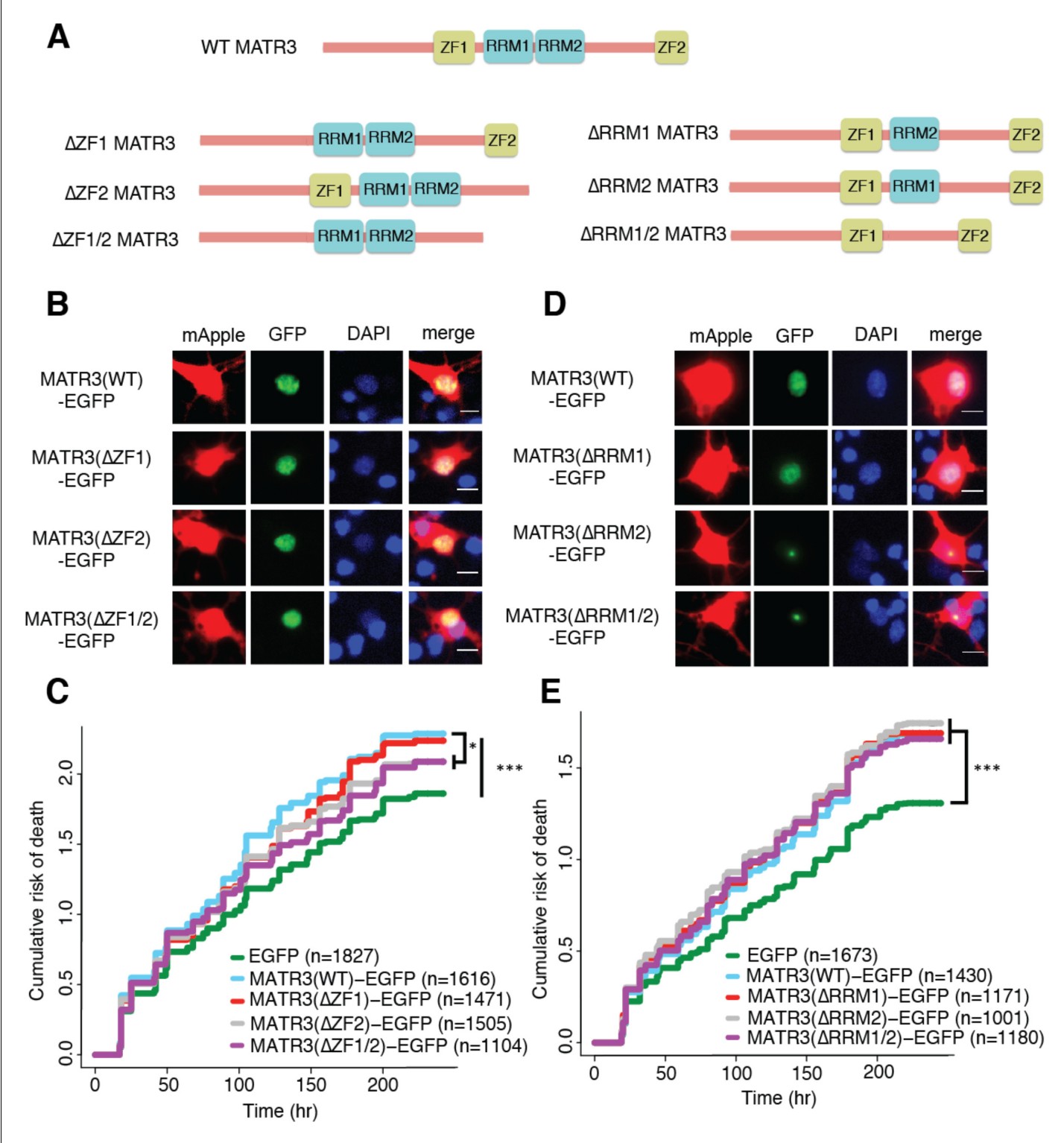

**Figure 3.** MATR3's ZFs mediate overexpression toxicity, and its RRMs regulate subcellular distribution. (**A**) Schematic of MATR3 domain deletion mutants. (**B**) Zinc finger (ZF) domain deletions do not change the localization of MATR3-EGFP compared to the full-length protein. (**C**) ZF2 deletion, either in isolation or combination with ZF1, results in modest rescue of overexpression toxicity (comparing to MATR3(WT)-EGFP, n = 1616: MATR3 (ΔZF1)-EGFP, HR = 0.94, n = 1471, p=0.10; MATR3(ΔZF2)-EGFP, HR = 0.93, n = 1505, *p=0.040; MATR3(ΔZF1/2)-EGFP, HR = 0.90, n = 1104, **p=0.0093; Cox proportional hazards). (**D**) While MATR3(ΔRRM1)-EGFP exhibits the same localization as MATR3(WT)-EGFP, deletion of RRM2 results in redistribution into intranuclear granules. (**E**) RRM deletion had little effect on MATR3-mediated toxicity (comparing to MATR3(WT)-EGFP n = 1430:
*Figure 3 continued on next page*

*Figure 3 continued*

MATR3(ΔRRM1)-EGFP, HR = 1.05, n = 1171, p=0.25; MATR3(ΔRRM2)-EGFP, HR = 1.09, n = 1001, p=0.066; MATR3(ΔRRM1/2)-EGFP, HR = 1.04, n = 1180, p=0.42). Scale bars in (**B**) and (**D**), 10 μm.

DOI: https://doi.org/10.7554/eLife.35977.006

The following source data and figure supplement are available for figure 3:

**Source data 1.** Survival data for *Figures 3C and E*.

DOI: https://doi.org/10.7554/eLife.35977.008

**Figure supplement 1.** MATR3(ΔRRM2)-EGFP and MATR3(ΔRRM1/2)-EGFP do not join preexisting subnuclear organelles.

DOI: https://doi.org/10.7554/eLife.35977.007

was 87.2% sensitive and 93.9% specific in discriminating cells with nuclear granules from those with diffuse protein (*Figure 4B*). We therefore utilized this CV threshold to assess the frequency of nuclear granule formation in primary rodent cortical neurons, noting that 24 hr after transfection, 76.1% (2081/2734) of neurons transfected with MATR3(ΔRRM2)-EGFP displayed nuclear granules compared to 91.2% (1590/1743) of MATR3(ΔRRM1/2)-EGFP cells (*Figure 4C*). We also observed the time-dependent formation of nuclear granules as neurons expressed increasing amounts of MATR3-EGFP (*Figure 4D*), suggesting that granule formation may be proportional to expression level. To investigate this relationship further, we identified neurons exhibiting a diffuse distribution of MATR3(ΔRRM2)-EGFP at day one and followed these cells for an additional 3 days by automated microscopy. We then measured the GFP intensity for each cell at day one and related this value to the risk of granule formation over the ensuing 72 hr period using penalized splines models. Notably, we failed to observe a significant relationship between GFP intensity on day one and granule formation by day 3 (*Figure 3E*). We also assessed the relative change in expression level on a per-cell basis, as quantified by the ratio of GFP intensity at day two to the GFP intensity at day 1, to determine if the net rate of MATR3(ΔRRM2)-EGFP production better predicted granule formation. The probability of granule formation was directly proportional to the time-dependent change in MATR3(ΔRRM2)-EGFP levels (*Figure 4F*), suggesting that granule formation is favored by the rapid accumulation of MATR3(ΔRRM2)-EGFP.

Our previous studies demonstrated that deletion of RRM1, RRM2, or both had no effect upon the toxicity of MATR3-EGFP when expressed in primary neurons (*Figure 3E*). These analyses included all neurons within a given condition, consisting of cells with diffuse nuclear MATR3 as well as those with MATR3 redistributed into granules. To determine if the presence of nuclear MATR3-EGFP granules impacted the survival of neurons, we utilized the nuclear CV threshold (*Figure 4B*) to divide neurons expressing MATR3(ΔRRM2)-EGFP and MATR3(ΔRRM1/2)-EGFP into three categories: cells with diffuse protein at day 1, those with granules at day 1, or all cells. We then tracked neurons in each category for the following 9 days by LFM, and compared their survival by Cox proportional hazards analysis. By these measures, neurons displaying nuclear MATR3(ΔRRM2)-EGFP granules fared significantly better than the population as a whole, while those exhibiting a diffuse distribution demonstrated an increased risk of death (*Figure 4G*). Similar results were obtained for neurons expressing MATR3(ΔRRM1/2)-EGFP; here, the relative protection associated with nuclear MATR3(ΔRRM1/2)-EGFP granules was modest, but the toxicity of diffusely-distributed MATR3(ΔRRM1/2)-EGFP was more pronounced (*Figure 4H*). The marked toxicity of diffuse MATR3(ΔRRM1/2)-EGFP may explain why so few cells with diffuse protein are seen at day 1 (*Figure 4D*). Taken together, these results suggest that diffuse MATR3 is highly neurotoxic when it cannot bind RNA. Furthermore, the sequestration of RNA binding-deficient MATR3 variants into nuclear granules is associated with a survival advantage.

## MATR3 granules formed by deletion of the RNA-binding domains display liquid-like properties that are affected by a pathogenic mutation

As part of their normal function, many RBPs reversibly undergo LLPS, involving the formation of droplets with liquid-like properties from diffuse or soluble proteins (*Molliex et al., 2015*; *Murray et al., 2017*). Disease-associated mutations in the genes encoding these proteins may promote LLPS or impair the reversibility of phase separation (*Kim et al., 2013*; *Molliex et al., 2015*;

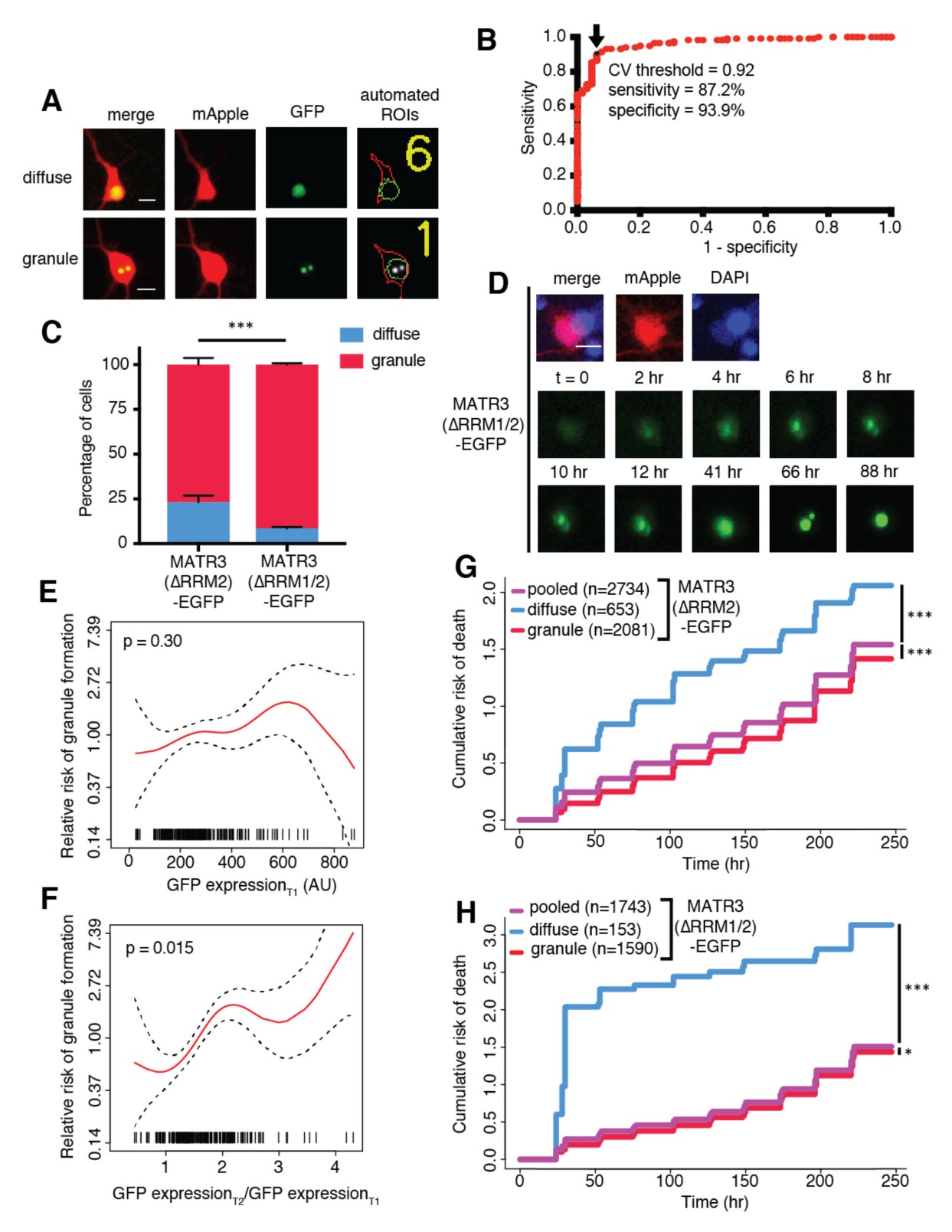

**Figure 4.** MATR3(ΔRRM2)-EGFP and MATR3(ΔRRM1/2)-EGFP are highly neurotoxic in their diffuse form. (**A**) Automated analysis of MATR3-EGFP distribution in transfected primary cortical neurons. Regions of interest (ROIs) were drawn around the cell body (marked by mApple fluorescence, red) and diffuse MATR3-EGFP (indicated by GFP fluorescence, green), and used to calculate a coefficient of variation (CV) representing MATR3-EGFP distribution within each ROI. (**B**) Receiver operating characteristic (ROC) curve for MATR3-EGFP CV values. A CV threshold of 0.92 (arrow/black point)

*Figure 4 continued on next page*

*Figure 4 continued*

identified cells with intranuclear MATR3-EGFP granules with 87.2% sensitivity and 93.9% specificity. (C) Using this cutoff, we determined that 1 day after transfection, 76.1% (2081/2734) of MATR3(ΔRRM2)-EGFP neurons displayed intranuclear granules compared to 91.2% (1590/1743) of MATR3(ΔRRM1/2)-EGFP cells (***p<0.00001, Fisher's exact test). (D) Intranuclear granules form in a time-dependent manner in neurons expressing MATR3(ΔRRM2)-EGFP and MATR3(ΔRRM1/2)-EGFP. (E–F) Penalized spline models depicting the relationship between MATR3(ΔRRM2)-EGFP expression on day 1 (E) or change in GFP expression between day 1 and day 2 (F), and risk of developing an intranuclear granule by day 3. Expression level at day one was not significantly associated with risk of granule formation (E; p=0.30, penalized spline regression), but the relative increase in expression from day 1 to day 2 is (F; p=0.015, penalized spline regression). (G) For MATR3(ΔRRM2)-EGFP, neurons exhibiting granules by day one displayed improved survival compared to the pooled combination of all cells. Conversely, neurons with diffusely distributed MATR3(ΔRRM2)-EGFP fared far worse (comparing to the pooled condition: cells with granules n = 2081, HR = 0.86, ***p=1.02 × 10$^{-5}$; cells with diffuse protein n = 653, HR = 1.75, ***p<2 × 10$^{-16}$; Cox proportional hazards). (H) Neurons with MATR3(ΔRRM1/2)-EGFP granules by day one similarly displayed a reduced risk of death in comparison to the pooled group, while diffuse MATR3(ΔRRM1/2)-EGFP was highly toxic (comparing to the pooled condition: cells with granules, n = 1590, HR = 0.92, *p=0.03; cells with diffuse protein, n = 153, HR = 3.78, ***p=2 × 10$^{-16}$; Cox proportional hazards). Scale bars in (A) and (B), 10 μm.

DOI: https://doi.org/10.7554/eLife.35977.009

The following source data is available for figure 4:

**Source data 1.** Sensitivity/specifity data for *Figure 4B*, granule formation risk and GFP expression data for *Figures 4E and F*, and survival data for *Figures 4G and H*.
DOI: https://doi.org/10.7554/eLife.35977.010

*Patel et al., 2015*; *Conicella et al., 2016*). We wondered whether the intranuclear granules formed by MATR3(ΔRRM2)-EGFP and MATR3(ΔRRM1/2)-EGFP represent liquid droplets and also whether pathogenic MATR3 mutations affect the intrinsic properties of these puncta. Indeed, nuclear granules exhibited dynamic properties, not only growing in size over time but also moving freely within the nucleus and fusing if they encountered other granules (*Figure 5A*), indicative of liquid-like behavior.

We then asked if these structures displayed internal rearrangement characteristic of liquid droplets (*Lin et al., 2015b*; *Shin and Brangwynne, 2017*) and whether pathogenic *MATR3* mutations affect their dynamics. To answer this, we introduced disease-associated mutations into MATR3(ΔRRM1/2)-EGFP and transfected rodent primary cortical neurons with each construct (*Figure 5B*). Nuclear puncta were photobleached 2–4 days after transfection, and the recovery of fluorescence intensity tracked within the bleached and unbleached ROIs by laser scanning confocal microscopy. Granules formed by MATR3(WT ΔRRM1/2)-EGFP displayed internal rearrangement over the course of minutes consistent with liquid-like properties, as did all tested disease mutants on the ΔRRM1/2 background (*Figure 5C–D*). The S85C mutation, however, severely slowed fluorescence recovery, suggesting reduced exchange of molecules within each droplet. Using the Stokes-Einstein equation, we calculated viscosity estimates for each MATR3(ΔRRM1/2)-EGFP variant based on return time and bleached area size (*Figure 5E*). Consistent with the observed effect of this mutation on fluorescence recovery, the S85C mutation led to a pronounced increase in viscosity over that of WT and other disease-associated mutants.

We wondered whether this phenotype was specific to nuclear droplets formed by MATR3(ΔRRM1/2)-EGFP, or if full-length MATR3 carrying pathogenic mutations would also display reduced mobility. For this, we transfected primary neurons with full-length versions of MATR3(WT)-EGFP or disease-associated MATR3-EGFP variants and then bleached a circular area in the center of the nucleus (*Figure 5F*). In each case, we noted rapid return of fluorescence, and the recovery rate was unaffected by pathogenic *MATR3* point mutations (*Figure 5G*). To account for the rapidity of return as well as the area of the bleached region, we calculated a diffusion coefficient (DC) for each construct. Comparison of the DCs for WT and mutant MATR3-EGFP variants showed no significant differences (*Figure 5H*). Our data therefore suggest that the S85C point mutation selectively affects the droplet properties of MATR3. To test whether this feature is shared among disease-associated mutations affecting the MATR3 N-terminus, we generated two additional pathogenic mutants, Q66K and A72T, on the ΔRRM1/2 background (*Figure 5—figure supplement 1*). Unlike the S85C variant, these mutations had no effect on granule viscosity, indicating that the S85C mutant is unique among N-terminal low-complexity domain mutations in its ability to affect the mobility of phase-separated MATR3.

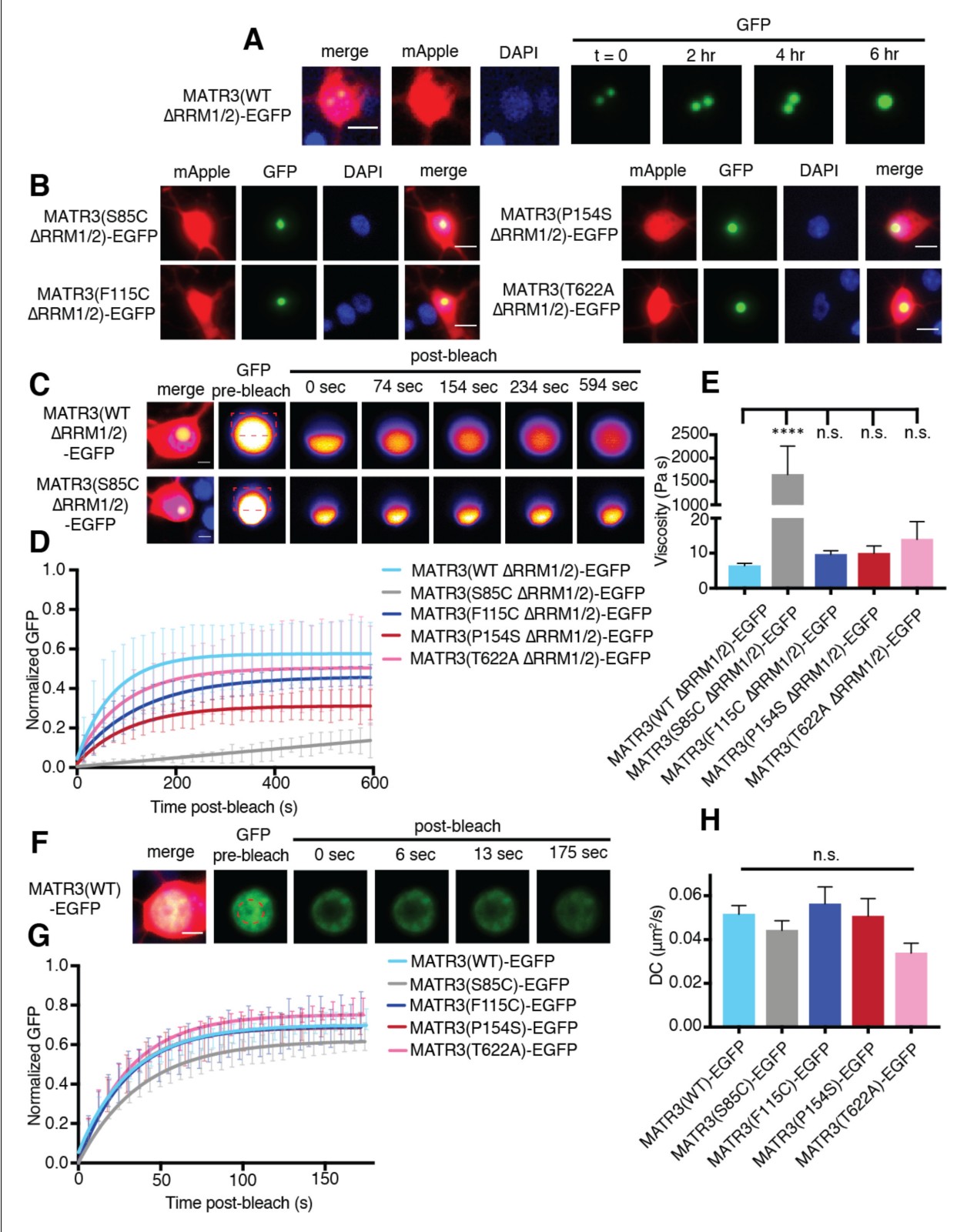

**Figure 5.** MATR3(ΔRRM1/2)-EGFP droplets display liquid-like properties that are affected by the S85C mutation. (**A**) MATR3(ΔRRM2)-EGFP and MATR3 (ΔRRM1/2)-EGFP droplets show liquid-like properties such as mobility and fusion. (**B**) Pathogenic MATR3 mutations on the ΔRRM1/2 background result in similar phase-separated droplets. (**C–D**) Fluorescence recovery after photobleaching (FRAP) of MATR3(ΔRRM1/2)-EGFP droplets shows internal rearrangement consistent with liquid-like behavior, but the recovery of MATR3(S85C ΔRRM1/2)-EGFP droplets was significantly delayed. (**E**) MATR3

*Figure 5 continued on next page*

*Figure 5 continued*

(S85C ΔRRM1/2)-EGFP droplets displayed significantly higher viscosity in comparison to MATR3(WT ΔRRM1/2)-EGFP (comparing to MATR3(WT ΔRRM1/2)-EGFP, n = 5: MATR3(S85C ΔRRM1/2)-EGFP, n = 5, ****p<0.0001; MATR3(F115C ΔRRM1/2)-EGFP, n = 5, p>0.9999; MATR3(P154S ΔRRM1/2)-EGFP, n = 5, p>0.9999; MATR3(T622A ΔRRM1/2)-EGFP, n = 4, p>0.9999; one-way ANOVA with Tukey's post-hoc test). (**F–G**) FRAP experiments involving full-length MATR3-EGFP variants showed no differences in rates of return. (**H**) Similarly, there were no differences in diffusion coefficients (DC) among full-length MATR3 variants (MATR3(WT)-EGFP, n = 5; MATR3(S85C)-EGFP, n = 5; MATR3(F115C)-EGFP; n = 5, MATR3(P154S)-EGFP, n = 5; MATR3(T622A)-EGFP, n = 4; p=0.17, one-way ANOVA). Scale bars in (**A**) and (**B**), 10 μm; scale bars in (**C**) and (**F**), 5 μm. Curves in (**D**) and (**G**) show fitted curves ± SD.
DOI: https://doi.org/10.7554/eLife.35977.011

The following figure supplement is available for figure 5:

**Figure supplement 1.** Pathogenic N-terminal domain mutations other than S85C on the ΔRRM1/2 background do not affect granule viscosity.
DOI: https://doi.org/10.7554/eLife.35977.012

## Mapping the sequence determinants of MATR3 localization in neurons

Cytoplasmic inclusions composed of the RBP TDP-43 are characteristic of ALS and the majority of FTD (*Arai et al., 2006*; *Neumann et al., 2006*). Moreover, pathogenic mutations in the gene encoding TDP-43 enhance cytoplasmic mislocalization concordant with enhanced neurotoxicity, and reductions in cytoplasmic TDP-43 prolong neuronal survival (*Barmada et al., 2010b*, *2014*). To determine if MATR3 localization is likewise an important determinant of neurodegeneration, we sought to disrupt the MATR3 nuclear localization signal (NLS). However, since multiple sequences have been associated with nuclear MATR3 localization (*Hibino et al., 2006*; *Hisada-Ishii et al., 2007*), we systematically identified regions enriched in positively-charged amino acids (arginine, lysine) that may mediate nuclear import via importin-α. We then deleted each of the seven regions defined in this manner, including two that had been identified as controlling nuclear localization in previous studies, and assessed their localization by transfection in rodent primary cortical neurons followed by fluorescence microscopy (*Figure 6A*).

Deletions of putative NLS (pNLS) 1, 2, 3, 5, 6, and 7 had little to no effect on neuronal MATR3 distribution (*Figure 6B*). While the ΔpNLS3 mutation did not change nuclear MATR3 localization per se, it did induce the formation of many small, nuclear granules. This effect is consistent with the position of pNLS3 within RRM2 and the observed formation of nuclear puncta upon RRM2 deletion (*Figure 4*). In contrast, and in accord with previous studies (*Hisada-Ishii et al., 2007*), deletion of the bipartite pNLS4 elicited a marked reduction in nuclear MATR3-EGFP accompanied by enhanced cytoplasmic localization and the formation of small MATR3-EGFP granules within the cytoplasm. In DT40 and HeLa cells, both arms of this NLS were critical for MATR3 nuclear localization (*Hisada-Ishii et al., 2007*). To determine if this is the case in neurons, we sequentially deleted the N- and C-terminal arms (ΔpNLS4N and ΔpNLS4C, respectively) and tested their localization by transfection into primary cortical neurons. These studies demonstrated that only the N-terminal arm is necessary for nuclear localization, as MATR3(ΔpNLS4N)-EGFP exhibits nuclear clearing and punctate distribution in the cytoplasm and neuronal processes, while MATR3(ΔpNLS4C)-EGFP has the same distribution as MATR3(WT)-EGFP (*Figure 6C–D*). To test whether pNLS4N was sufficient for nuclear localization, we generated a construct in which the eight amino acids corresponding to pNLS4N were appended to EGFP, and compared the subcellular distribution of this construct in primary neurons to EGFP alone or EGFP fused to the canonical NLS from the SV40 large T antigen (*Figure 6—figure supplement 1*) (*Kalderon et al., 1984*). These studies demonstrated comparable nuclear localization of pNLS4N-EGFP and SV40 NLS-EGFP, indicating that the MATR3 pNLS4N sequence is both necessary and sufficient for nuclear protein localization in primary neurons.

Having identified the N-terminal arm of the bipartite pNLS4 as the key sequence regulating MATR3 localization in neurons, we asked whether driving MATR3 into the cytoplasm by deletion of this sequence could modify toxicity. Rodent primary cortical neurons were transfected with mApple and either EGFP, MATR3(WT)-EGFP, or MATR3(ΔpNLS4N)-EGFP and imaged at regular intervals by LFM. Automated survival analysis of neuronal populations expressing these constructs demonstrated that the ΔpNLS4N mutation and resulting cytoplasmic localization significantly reduced MATR3-dependent toxicity compared to the MATR3(WT)-EGFP (*Figure 6E*). Therefore, unlike TDP-43 and FUS, two RBPs whose cytoplasmic mislocalization is tightly tied to neurodegeneration in ALS/FTD models, cytoplasmic MATR3 retention mitigates toxicity, suggesting that nuclear MATR3 functions are required for neurodegeneration (*Barmada et al., 2010b*; *Qiu et al., 2014*).

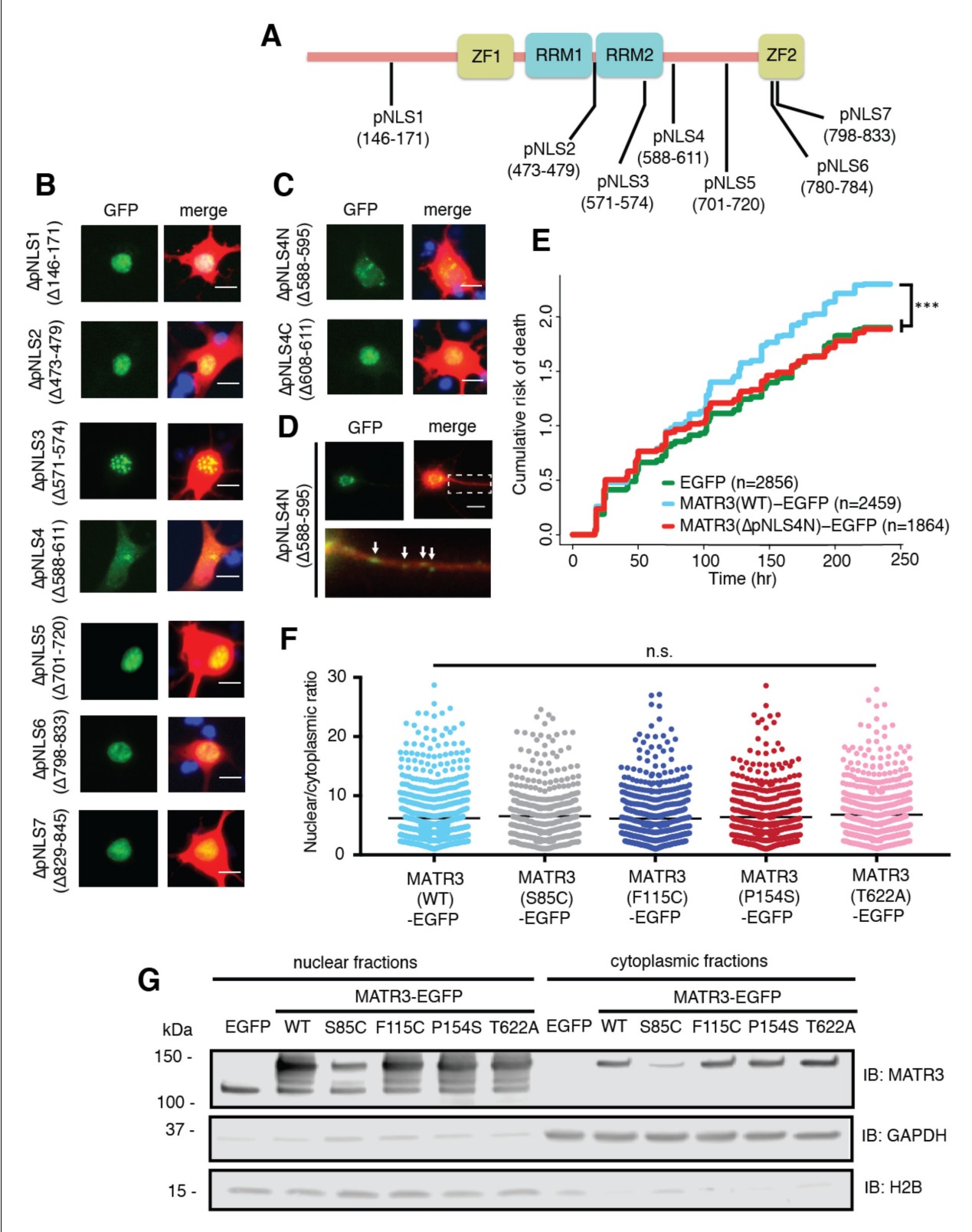

**Figure 6.** Reducing MATR3 nuclear localization mitigates overexpression toxicity. (A) Schematic showing putative MATR3 nuclear localization signals (pNLS). (B–C) Deletion of the N-terminal arm of NLS4 (ΔpNLS4N) led to nuclear MATR3 clearance in neurons. (D) MATR3(ΔpNLS4N)-EGFP forms granular structures in the cytoplasm and neuronal processes (white arrows). (E) Disrupting nuclear localization of MATR3 prevents neurotoxicity from overexpression (compared to MATR3(WT)-EGFP, n = 2459: MATR3(ΔpNLS4N)-EGFP, n = 1864, HR = 0.89, ***p=0.00041, Cox proportional hazards). (F–

*Figure 6 continued on next page*

Figure 6 continued

G) Pathogenic MATR3 mutants display no difference in subcellular protein localization as assessed by automated image nuclear/cytoplasmic analysis (F; MATR3(WT)-EGFP, n = 789; MATR3(S85C)-EGFP, n = 462; MATR3(F115C)-EGFP, n = 596; MATR3(P154S)-EGFP, n = 524; MATR3(T622A)-EGFP, n = 657; p=0.067, one-way ANOVA) or biochemical fractionation in transfected HEK293T cells (G). Western blot demonstrated reduced abundance of the S85C mutant in transfected HEK293T cells. Scale bars in (B) and (C), 10 μm; scale bar in (D), 50 μm.
DOI: https://doi.org/10.7554/eLife.35977.013

The following source data and figure supplement are available for figure 6:

Source data 1. Survival data for *Figure 6E* and nucleocytoplasmic localization data for *Figure 6F* and *Figure 6—figure supplement 1B*.
DOI: https://doi.org/10.7554/eLife.35977.015

Figure supplement 1. The N-terminal arm of MATR3's bipartite NLS is capable of driving nuclear enrichment of a heterologous protein.
DOI: https://doi.org/10.7554/eLife.35977.014

Given the observed relationship between MATR3 localization and toxicity, we wondered if subtle changes in nucleocytoplasmic MATR3 distribution could be responsible for the increased toxicity of MATR3 bearing disease-associated mutations. Rodent primary cortical neurons transfected with each of the pathogenic MATR3-EGFP variants showed no obvious difference in subcellular localization in comparison with MATR3(WT)-EGFP (*Figure 2A*). To investigate MATR3-EGFP localization in a quantitative manner, we developed a customized image-based analysis script (https://github.com/barmadaslab/nuclear-fractionation; *Miguez, 2018c*; copy archived at https://github.com/elifesciences-publications/nuclear-fractionation) to draw ROIs around the nucleus and soma of each neuron, measure MATR3-EGFP content separately within each compartment, and calculate a nucleocytoplasmic ratio for MATR3-EGFP in individual cells (*Figure 6F*). This analysis confirmed our initial observations, showing no significant differences in the localization of mutant MATR3-EGFP variants compared to MATR3(WT)-EGFP.

In a complementary series of experiments, we utilized biochemical fractionation to assess the distribution of MATR3-EGFP in a human cell line. MATR3(WT)-EGFP or versions of MATR3-EGFP bearing the S85C, F115C, P154S, and T622A disease-associated mutations were transfected into HEK293T cells, and the nuclear and cytoplasmic fractions subjected to SDS-PAGE and Western blotting. In agreement with single-cell data from transfected primary neurons, we noted no difference in the nucleocytoplasmic distribution of any of the MATR3-EGFP variants tested here (*Figure 6G*). Nevertheless, we consistently observed far less of the S85C variant in both nuclear and cytoplasmic fractions, compared to MATR3(WT)-EGFP and other disease-associated mutants. These data suggest that the S85C mutation may destabilize MATR3-EGFP; alternatively, this mutation may prevent adequate solubilization and detection of MATR3-EGFP via SDS-PAGE and Western blotting.

## A subset of pathogenic MATR3 mutations affect protein solubility and stability

To discriminate among these possibilities, we first investigated the turnover of WT and mutant MATR3 variants using optical pulse labeling (OPL), a technique enabling non-invasive determinations of protein clearance in living cells (*Barmada et al., 2014*). For these experiments, MATR3 was fused to Dendra2—a photoconvertible protein that irreversibly switches from a green to red fluorescent state upon illumination with low-wavelength light (*Chudakov et al., 2007*)—and expressed in primary cortical neurons. One day after transfection, neurons were illuminated with blue light to photoconvert Dendra2, and the time-dependent loss of red fluorescence signal used to calculate protein half-life (*Figure 7A*). Previous studies validated the accuracy and utility of OPL for determinations of protein half-life (*Barmada et al., 2014*); importantly, and in contrast to biochemical techniques for calculating half-life that depend on radioactive labeling or translational inhibitors, OPL allows us to measure protein clearance on a single-cell level for thousands of neurons simultaneously (*Figure 7B*). Most disease-associated mutations had no effect upon the turnover of MATR3-Dendra2 in primary cortical neurons. However, we noted subtle destabilization of MATR3(S85C)-Dendra2 in comparison to other pathogenic mutant variants and MATR3(WT)-Dendra2 (*Figure 7C–D*). Even so, the magnitude of the effect was relatively small, making it unlikely that differences in protein turnover fully explain the reduced abundance of MATR3(S85C)-EGFP noted in cell lysates (*Figure 6G*).

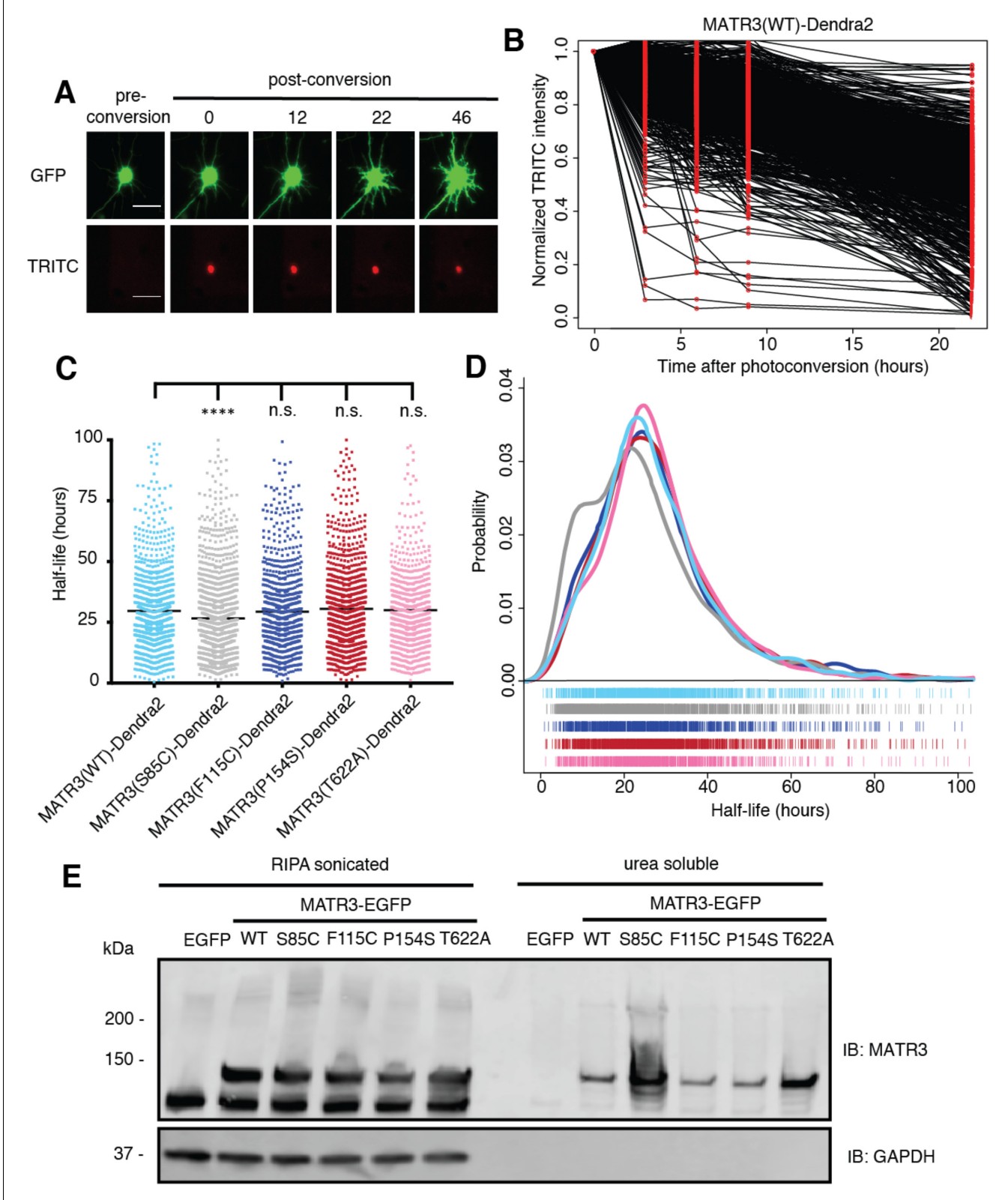

**Figure 7.** Pathogenic MATR3 mutations have little effect on MATR3 turnover, but a subset reduce solubility. (A) Optical pulse labeling of Dendra2-tagged MATR3 variants. Each neuron is transfected with EGFP alone to outline the cell body, as well as MATR3-Dendra2, which fluoresces in the red channel (TRITC) upon photoconversion. Scale bar, 50 μm. (B) Normalized red fluorescence (TRITC) signal for individual neurons. The time-dependent decay of red fluorescence over time is used to calculate MATR3-Dendra2 half-life for each neuron. (C–D) MATR3(S85C)-Dendra2 displayed a subtle but

*Figure 7 continued on next page*

*Figure 7 continued*

significant reduction in half-life compared to MATR3(WT)-Dendra2 (comparing to MATR3(WT)-Dendra2, n = 1269: MATR3(S85C)-Dendra2, n = 1670, ****p<0.0001; MATR3(F115C)-Dendra2, n = 1122, p>0.9999; MATR3(P154S)-Dendra2, n = 1509, p=0.9309; MATR3(T622A)-Dendra2, n = 923, p=0.9989; one-way ANOVA with Tukey's post-hoc test). (E) Sonication in RIPA resulted in equivalent amounts of all MATR3 variants by Western blotting. The S85C variant was markedly enriched in the RIPA-insoluble, urea-soluble fraction, while the T622A variant showed more modest enrichment.

DOI: https://doi.org/10.7554/eLife.35977.016

The following source data is available for figure 7:

**Source data 1.** Half-life data for *Figures 7C and D*.

DOI: https://doi.org/10.7554/eLife.35977.017

We next asked if the S85C mutation altered MATR3 solubility. HEK293T cells transfected with WT and mutant MATR3-EGFP variants were lysed using a harsher protocol that involved sonication in RIPA buffer; additionally, we used urea buffer to extract all RIPA-insoluble proteins. In stark contrast to mild conditions (*Figure 6G*), harsher lysis resulted in equivalent levels of all MATR3 variants on Western blot, suggesting that the S85C mutation reduced MATR3 solubility (*Figure 7E*). In accordance with this interpretation, the urea-soluble fraction was markedly enriched for MATR3(S85C)-EGFP and modestly enriched for MATR3(T622A)-EGFP. These data show that the S85C and T622A mutations reduce the solubility of MATR3, without drastically affecting its stability. As shown in *Figure 1A*, both mutations lie within areas of predicted disorder, consistent with their effects on MATR3 aggregation and solubility.

## Discussion

In this study, we modeled MATR3-mediated neurodegeneration by overexpressing WT or disease-associated MATR3 variants in primary neurons. In doing so, we found that neurons were highly susceptible to increases or decreases in MATR3 levels, and disease-associated MATR3 variants exhibited enhanced toxicity in comparison to MATR3(WT). Structure-function studies demonstrated that the ZF2 domain modulates overexpression-related toxicity, while RRM2 prevents MATR3 phase separation into mobile nuclear puncta. Biophysical analysis of these puncta confirmed their liquid-like nature and further indicated that the pathogenic S85C mutation substantially increased the viscosity of these structures. We also determined that the N-terminal arm of a bipartite NLS drives MATR3 nuclear localization in neurons; forcing MATR3 into the cytoplasm by deleting this sequence blocked toxicity from MATR3 overexpression. While we did not observe any differences in the distribution of pathogenic MATR3 variants, we noted that the S85C mutation significantly reduced MATR3 solubility and, to a lesser extent, stability. The T622A mutant displayed similar but more muted effects on MATR3 solubility, suggesting that disease-associated mutations located in distinct MATR3 domains may operate through convergent pathogenic mechanisms.

Both MATR3 overexpression and knockdown elicited significant and comparable toxicity in neurons. These data suggest that neurons are bidirectionally vulnerable to changes in MATR3 levels. Post-mortem studies of MATR3 distribution in sporadic and familial ALS patients demonstrated stronger MATR3 nuclear staining as well as the presence of cytoplasmic MATR3 aggregates in motor neurons (*Dreser et al., 2017*; *Tada et al., 2018*). While the impact of these findings is unknown, MATR3 mislocalization or sequestration into aggregates may reflect a reduction in normal function, a new and abnormal function, or both. In mice, homozygous *Matr3* knockout is embryonic lethal, while heterozygous *Matr3*[+/-] animals demonstrate incompletely penetrant cardiac developmental abnormalities. However, *Matr3*[+/-] mice exhibited roughly equivalent Matr3 protein levels in comparison to nontransgenic animals, complicating any conclusions regarding Matr3 loss-of-function in these models (*Quintero-Rivera et al., 2015*). Overexpression of human MATR3(F115C) in mice results in severe muscle disease consisting of fore- and hindlimb muscle atrophy accompanied by vacuolization (*Moloney et al., 2016*). These animals also displayed spinal cord gliosis and cytoplasmic MATR3 redistribution in spinal motor neurons akin to changes in MATR3 localization noted in humans with ALS, although no significant neurodegeneration was observed in MATR3(F115C) transgenic mice. Our data illustrating the dose-dependency of MATR3 neurotoxicity (*Figure 1*) imply that MATR3 (F115C) expression may be insufficient to elicit neurodegeneration in these animals. Alternatively,

constitutive overexpression of MATR3(F115C) in transgenic mice may trigger compensatory mechanisms during development that promote neuronal survival.

MATR3 is unique among ALS/FTD-associated RBPs in possessing not just two tandem RRMs but also two ZF domains that can bind repetitive DNA elements found in the nuclear scaffold (*Hibino et al., 1998*). MATR3 binds thousands of RNAs via a pyrimidine-rich consensus sequence (UUUCUXUUU; *Uemura et al., 2017*); these binding events are concentrated within introns and most often associated with exon repression. We found that genetic ablation of either or both of MATR3's RRMs had no overall effect on overexpression-mediated toxicity. Conversely, ZF2 deletion mitigated MATR3 overexpression-mediated toxicity, suggesting that aberrant DNA binding by overexpressed MATR3 may be partially responsible for neurodegeneration in these systems. MATR3's genomic targets remain uncharacterized, however, and further studies are required to identify relevant MATR3 DNA substrates that participate in MATR3 overexpression-related toxicity.

Our data support a model in which RNA binding prevents MATR3 self-association into droplets. Consistent with this interpretation, deletion of RRM2—either alone or in combination with RRM1—resulted in the formation of phase-separated intranuclear droplets. We also observed small, mobile MATR3 granules in the cytoplasm and neuronal processes when the bipartite NLS was disrupted (*Figure 6D*). Cytoplasmic RNA concentrations are more than an order of magnitude lower than those in the nucleus, a gradient that may favor the coalescence of MATR3(ΔpNLS4N)-EGFP into puncta within the neuronal soma and processes (*Goldstein and Trescott, 1970*). Similarly, phase transitions of two other RBPs implicated in ALS and FTD—TDP-43 and FUS—are blocked by high RNA concentrations in the nucleus and facilitated by low RNA concentrations in the cytoplasm (*Maharana et al., 2018*), indicating that this phenomenon is not exclusive to MATR3.

In C2C12 myoblast cells, MATR3 formed intranuclear spherical structures with liquid-like properties upon deletion of RRM2, though these granules were smaller and more numerous than those we detected in primary neurons, a difference that may be due to expression level and cell type (*Iradi et al., 2018*). RRM2 deletion in C2C12 cells led to a large increase in protein binding partners, many with low-complexity domains. In the absence of effective RNA binding, therefore, MATR3 may be free to interact with other proteins and itself through its low-complexity domains, driving LLPS.

The functional importance of the individual RRM domains for MATR3's RNA binding activity is unclear; while some studies suggest that both RRM1 and RRM2 bind RNA, other investigations indicated that RRM2 is primarily responsible for RNA binding (*Hibino et al., 2006*; *Salton et al., 2011*). Our data show that deletion of RRM2 is sufficient to elicit MATR3 phase separation, suggesting that RNA recognition by MATR3 is mediated largely by RRM2. We also noted no significant difference in the survival of neuronal populations overexpressing ΔRRM1, ΔRRM2, and ΔRRM1/2 variants of MATR3-EGFP, implying that RNA binding per se is unrelated to MATR3-mediated neurodegeneration. This interpretation is strengthened by detailed analyses of neurons expressing MATR3(ΔRRM2) and MATR3(ΔRRM1/2). When neurons with and without droplets were assessed separately, we noted that neurons exhibiting diffuse MATR3(ΔRRM2) or MATR3(ΔRRM1/2) displayed a significantly higher risk of death than those with droplets. These results imply that diffuse MATR3, when not bound to RNA, can be highly toxic. Conversely, sequestration of RNA-binding deficient MATR3 into puncta is associated with extended neuronal survival. Our data further indicate that diffuse MATR3(ΔRRM1/2) is more toxic than diffuse MATR3(ΔRRM2) (compare the diffuse population in *Figure 4G* to the diffuse population in *Figure 4H*). Since RRM1 may be capable of recognizing some RNA even without RRM2, these observations suggest that neurodegeneration is inversely proportional to the ability of MATR3 to bind RNA when diffusely localized within the nucleus. In disease models involving related RBPs, including TDP-43 and FUS, toxicity requires the presence of RNA binding motifs as well as low-complexity domains that enable LLPS (*Johnson et al., 2008*; *Daigle et al., 2013*; *Ihara et al., 2013*). As with MATR3, abrogation of RNA binding may disinhibit self-association, resulting in the sequestration of otherwise toxic diffuse protein within droplets.

Investigating the liquid-like properties of MATR3(ΔRRM1/2)-EGFP droplets, we noted a selective effect of the S85C mutation on droplet viscosity. Low-complexity, intrinsically disordered domains are required for phase separation and self-assembly of RBPs. Apart from its nucleic acid binding domains, MATR3 displays a high degree of predicted disorder based on its primary amino acid sequence (*Figure 1A*). Among the pathogenic mutations studied here, only the S85C mutation

significantly affected MATR3(ΔRRM2)-EGFP droplet viscosity; notably, S85C is also the only disease-associated mutation associated with a primary myopathy as well as neurodegeneration (*Feit et al., 1998*; *Senderek et al., 2009*). In myoblasts, MATR3(ΔRRM2) carrying the S85C mutation did not form spherical droplets but rather coalesced into irregular nuclear clusters, pointing to cell type-specific differences in MATR3 behavior that may be relevant for myopathic or neurodegenerative phenotypes (*Iradi et al., 2018*). Whether full-length MATR3 is capable of phase separation under physiological circumstances, and what relevance this process has for disease, is currently unclear.

Conflicting evidence (*Hibino et al., 2006*; *Hisada-Ishii et al., 2007*) suggests that MATR3 nuclear import is driven by distinct sequences in different cell types. For example, while amino acids 701–718 are essential for nuclear localization of rat MATR3 in Ac2F cells, deletion of the homologous sequence (amino acids 701–720) in human MATR3 has no effect on neuronal distribution (*Figure 6B*). To identify the sequences responsible for MATR3 nuclear import within neurons, we undertook a systematic analysis of arginine/lysine-rich sequences in MATR3 resembling NLSs. In accord with an earlier report (*Hisada-Ishii et al., 2007*), we found that MATR3's bipartite pNLS controlled its nuclear enrichment in neurons, but only the N-terminal arm of this pNLS was necessary and sufficient for MATR3 nuclear localization in neurons. Pathogenic *TARDBP* and *FUS* mutations promote cytoplasmic mislocalization of TDP-43 and FUS, respectively, and cytoplasmic enrichment of these proteins is tightly linked to toxicity (*Barmada et al., 2010b*; *Dormann et al., 2010*). In stark contrast, however, we observed that cytoplasmic MATR3 redistribution extended neuronal survival, suggesting—along with the partial rescue we observed for MATR3(ΔZF2)-EGFP and MATR3(ΔZF1/2)-EGFP—that MATR3 overexpression elicits neurodegeneration through nuclear DNA binding activity, mediated at least in part by ZF2.

Given previously established relationships between the distribution and aggregation of RBPs and neurodegeneration in ALS models (*Johnson et al., 2009*; *Barmada et al., 2010b*; *Dormann et al., 2010*; *Igaz et al., 2011*; *Kim et al., 2013*; *Qiu et al., 2014*), we wondered whether the enhanced toxicity of pathogenic MATR3 variants arises from mutation-associated changes in MATR3 localization or solubility. We noted no significant differences in the subcellular distribution of mutant MATR3 variants in comparison to MATR3(WT) but instead consistently observed reduced levels of MATR3(S85C) in transfected cell lysates. A similar pattern was noted in previous investigations and attributed to reduced MATR3(S85C) stability (*Johnson et al., 2014*). Using OPL, a sensitive method for measuring protein turnover in situ (*Barmada et al., 2014*; *Gupta et al., 2017*), we detected only a very modest shortening of MATR3(S85C) half-life compared to MATR3(WT). Nevertheless, we observed a marked change in the solubility of MATR3(S85C) and, less so, MATR3(T622A). This is in partial agreement with initial studies of MATR3(S85C) in patient tissues that noted equivalent amounts of MATR3(WT) and MATR3(S85C) in insoluble fractions but reduced MATR3(S85C) in nuclear fractions (*Senderek et al., 2009*). Both the S85C and T622A mutations lie within domains predicted to be disordered (*Figure 1A*). Furthermore, both mutations disrupt potential phosphorylation sites, and phosphorylation within the intrinsically disordered domain of FUS inhibits self-association of the protein through negative-negative charge repulsion between phosphate groups (*Monahan et al., 2017*). Of the 13 pathogenic mutations identified to date in MATR3, four (S85C, S610F, T622A, S707L) eliminate phosphorylatable residues, suggesting that inadequate phosphorylation and subsequent disinhibited self-association of MATR3 may be a conserved feature of MATR3 mutants.

MATR3's possesses broad functions in DNA/RNA processing (*Belgrader et al., 1991*; *Hibino et al., 2000*; *Zhang and Carmichael, 2001*; *Salton et al., 2010*; *Coelho et al., 2015*; *Rajor et al., 2016*; *Uemura et al., 2017*). Its presence within cytoplasmic aggregates in approximately half of patients with sporadic ALS (*Tada et al., 2018*) implies that MATR3 pathology causes or is caused by cellular alterations in RNA and protein homeostasis, many of which may contribute to neurodegeneration in ALS and related disorders. Our work confirms that MATR3 is essential for maintaining neuronal survival and furthermore shows that MATR3 accumulation results in neurodegeneration in a manner that depends on its subcellular localization and ZF domains. Additional studies are needed to further delineate the impact of disease-associated MATR3 mutations on the function, behavior, and liquid-like properties of MATR3.

# Materials and methods

## Key resources table

| Reagent type | Designation | Source | Identifiers | Additional information |
|---|---|---|---|---|
| Cell line (*Homo sapiens*) | HEK293T | ATCC | CRL-3216; RRID:CVCL_0063 | |
| Antibody | Rabbit anti-MATR3 | Abcam | EPR10635(B); RRID:AB_2491618 | (1:1000) for ICC in *Figure 2* and Western blot in *Figure 6* and 7 |
| Antibody | Rabbit anti-MATR3 | Abcam | EPR10634(B) | (1:1000) for ICC in *Figure 1* |
| Antibody | Mouse anti-fibrillarin | Abcam | 38F3; RRID:AB_304523 | (1:1000) |
| Antibody | Mouse anti-SC35 | Novus Biologicals | NB100-1774SS; RRID:AB_526734 | (1:2000) |
| Antibody | Mouse anti-PML | Santa Cruz Biotechnology | sc-377390 | (1:50) |
| Antibody | Goat anti-mouse 647 | Jackson Immuno Research | 115-605-003; RRID:AB_2338902 | (1:1000) |
| Antibody | Goat anti-rabbit 647 | Jackson Immuno Research | 111-605-003; RRID:AB_2338072 | (1:1000) |
| Antibody | Mouse anti-GAPDH | Millipore Sigma | MAB374; RRID:AB_2107445 | (1:1000) |
| Antibody | Rabbit anti-H2B | Novus Biologicals | NB100-56347; RRID:AB_838347 | (1:1000) |
| Antibody | AlexaFluor goat anti-mouse 594 | ThermoFisher | A-11005; RRID:AB_141372 | (1:10,000) |
| Antibody | AlexaFluor goat anti-rabbit 488 | ThermoFisher | A-11008; RRID:AB_143165 | (1:10,000) |
| Recombinant DNA reagent | MATR3 cDNA | Addgene | #32880 | |
| Recombinant DNA reagent | pGW1 MATR3(WT)-EGFP | This paper | | |
| Recombinant DNA reagent | pGW1 MATR3(S85C)-EGFP | This paper | | |
| Recombinant DNA reagent | pGW1 MATR3(F115C)-EGFP | This paper | | |
| Recombinant DNA reagent | pGW1 MATR3(P154S)-EGFP | This paper | | |
| Recombinant DNA reagent | pGW1 MATR3(T622A)-EGFP | This paper | | |
| Recombinant DNA reagent | pGW1 MATR3(ΔZF1)-EGFP | This paper | | |
| Recombinant DNA reagent | pGW1 MATR3(ΔZF2)-EGFP | This paper | | |
| Recombinant DNA reagent | pGW1 MATR3(ΔZF1/2)-EGFP | This paper | | |
| Recombinant DNA reagent | pGW1 MATR3(ΔRRM1)-EGFP | This paper | | |
| Recombinant DNA reagent | pGW1 MATR3(ΔRRM2)-EGFP | This paper | | |
| Recombinant DNA reagent | pGW1 MATR3(ΔRRM1/2)-EGFP | This paper | | |
| Recombinant DNA reagent | pGW1 MATR3(Q66K ΔRRM1/2)-EGFP | This paper | | |
| Recombinant DNA reagent | pGW1 MATR3(A72T ΔRRM1/2)-EGFP | This paper | | |
| Recombinant DNA reagent | pGW1 MATR3(S85C ΔRRM1/2)-EGFP | This paper | | |
| Recombinant DNA reagent | pGW1 MATR3(F115C ΔRRM1/2)-EGFP | This paper | | |

*Continued on next page*

*Continued*

| Reagent type | Designation | Source | Identifiers | Additional information |
|---|---|---|---|---|
| Recombinant DNA reagent | pGW1 MATR3(P154S ΔRRM1/2)-EGFP | This paper | | |
| Recombinant DNA reagent | pGW1 MATR3(T622A ΔRRM1/2)-EGFP | This paper | | |
| Recombinant DNA reagent | pGW1 MATR3(ΔpNLS1)-EGFP | This paper | | |
| Recombinant DNA reagent | pGW1 MATR3(ΔpNLS2)-EGFP | This paper | | |
| Recombinant DNA reagent | pGW1 MATR3(ΔpNLS3)-EGFP | This paper | | |
| Recombinant DNA reagent | pGW1 MATR3(ΔpNLS4)-EGFP | This paper | | |
| Recombinant DNA reagent | pGW1 MATR3(ΔpNLS5)-EGFP | This paper | | |
| Recombinant DNA reagent | pGW1 MATR3(ΔpNLS6)-EGFP | This paper | | |
| Recombinant DNA reagent | pGW1 MATR3(ΔpNLS7)-EGFP | This paper | | |
| Recombinant DNA reagent | pGW1 MATR3(ΔpNLS4N)-EGFP | This paper | | |
| Recombinant DNA reagent | pGW1 MATR3(ΔpNLS4C)-EGFP | This paper | | |
| Recombinant DNA reagent | pGW1 pNLS4N-EGFP | This paper | | |
| Recombinant DNA reagent | pGW1 SV40 large T antigen NLS-EGFP | This paper | | |
| Recombinant DNA reagent | pGW1 MATR3(WT)-Dendra2 | This paper | | |
| Recombinant DNA reagent | pGW1 MATR3(S85C)-Dendra2 | This paper | | |
| Recombinant DNA reagent | pGW1 MATR3(F115C)-Dendra2 | This paper | | |
| Recombinant DNA reagent | pGW1 MATR3(P154S)-Dendra2 | This paper | | |
| Recombinant DNA reagent | pGW1 MATR3(T622A)-Dendra2 | This paper | | |

## Plasmids

Full-length human *MATR3* cDNA was obtained from Addgene (#32880) and cloned into the pCMV-Tag2B vector (Agilent Technologies, #211172, Santa Clara, CA) using BamHI and XhoI endonucleases, tagging the amino-terminus with a FLAG epitope. To generate MATR3-EGFP, the *EGFP* open reading frame with a 14 amino acid N-terminal linker was amplified from pGW1-EGFP (*Arrasate et al., 2004*) by PCR using forward primer 5'-AGC TAC TAG TAC TAG AGC TGT TTG GGA C-3' and reverse primer 5'-TAT TGG GCC CCT ATT ACT GTT ACA GCT CGT CCA T-3'. The resulting amplicon was digested with SpeI and ApaI and cloned into the corresponding sites in pKS to generate pKS-EGFP. To create pKS-MATR3-EGFP, the *FLAG-MATR3* open reading frame from pCMV-Tag2B was amplified by PCR with forward primer 5'-GAT CTC TAG AGC GGC CGC CAC CAT GGA T-3' and reverse primer 5'-AGC TAC TAG TCA TAG TTT CCT TCT TCT GTC T-3', digested with XbaI and SpeI, and inserted into the corresponding sites in pKS-EGFP. pGW1-MATR3-EGFP was generated by digesting pKS-MATR3-EGFP with XbaI and ApaI, purifying the ensuing fragment containing MATR3-EGFP, and inserting into the corresponding sites of pGW1. To create Dendra2-tagged MATR3 variants, the *EGFP* coding region of each construct was removed by PCR amplification of the pGW1-MATR3-EGFP vector using primers that flank the *EGFP* open reading frame. The *Dendra2* open reading frame was then removed from pGW1-Dendra2 (*Barmada et al.,*

*2014*) by digestion with ApaI and MfeI, and inserted into pGW1-MATR3. All constructs were confirmed by sequencing prior to transfection in neurons and HEK293T cells.

Domain deletion mutants were created using Q5 Hot Start High-Fidelity DNA Polymerase (New England Biolabs, Ipswich, MA) and primers flanking the regions to be deleted for nucleic acid-binding domain (*Table 1*) and putative nuclear localization signal (*Table 2*) deletions. All disease-associated point mutations were created with site-directed mutagenesis (*Table 3*).

To generate NLS-EGFP fusion protein variants, the pNLS4N sequence (5'-AAA AAA GAT AAA TCC CGA AAA AGA-3') and SV40 large T antigen NLS sequence (5'-CCA AAA AAG AAG AGA AAG GTA-3') were synthesized as oligonucleotides flanked with ends complementary to ApaI/AgeI sites and BsrGI/EcoRI sites, respectively. These were heated to 95°C for 5 min, annealed overnight at room temperature (RT), and phosphorylated with T4 Polynucleotide Kinase (New England Biolabs). pGW1-EGFP and FUGW-EGFP vectors were digested with ApaI/AgeI and BsrGI/EcoRI, respectively, after which the corresponding annealed NLS-containing oligonucleotides were ligated.

## Primary neuron cell culture and transfection

Cortices from embryonic day (E)19–20 Long-Evans rat embryos were dissected and disassociated, and primary neurons plated at a density of $6 \times 10^5$ cells/mL in 96-well plates, as described previously (*Saudou et al., 1998*). At in vitro day (DIV) 4–5, neurons were transfected with 100 ng of pGW1-mApple (*Barmada et al., 2014*) to mark cells bodies and 100 ng of an experimental construct (i.e. pGW1-MATR3-EGFP) using Lipofectamine 2000, as before (*Barmada et al., 2010b*). Following transfection, cells were placed into either Neurobasal with B27 supplement (Gibco, Waltham, MA; for all survival experiments) or NEUMO photostable medium (Cell Guidance Systems, Cambridge, UK; for optical pulse labeling experiments). For siRNA knockdown experiments, neurons were transfected with 100 ng of pGW1-mApple per well and siRNA at a final concentration of 90 nM. Cells were treated with either scrambled siRNA (Dharmacon, Lafayette, CO) or siRNA targeting the N-terminal coding region of rat Matr3 (5'-GUC AUU CCA GCA GUC AUC UUU-3').

## Longitudinal fluorescence microscopy and automated image analysis

Neurons were imaged as described previously (*Barmada et al., 2015*) using a Nikon (Tokyo, Japan) Eclipse Ti inverted microscope with PerfectFocus3 and a 20X objective lens. Detection was accomplished with an Andor (Belfast, UK) iXon3 897 EMCCD camera or Andor Zyla4.2 (+) sCMOS camera. A Lambda XL Xenon lamp (Sutter) with 5 mm liquid light guide (Sutter Instrument, Novato, CA) was used to illuminate samples, and custom scripts written in Beanshell for use in µManager controlled all stage movements, shutters, and filters. Custom ImageJ/Fiji macros and Python scripts (https://github.com/barmadaslab/survival-analysis and https://github.com/barmadaslab/measurements; *Miguez, 2018a*; *Miguez, 2018b*; copies archived at https://github.com/elifesciences-publications/survival-analysis and https://github.com/elifesciences-publications/measurements) were used to identify neurons and draw regions of interest (ROIs) based upon size, morphology, and fluorescence intensity. Criteria for marking cell death involved rounding of the soma, loss of fluorescence and degeneration of neuritic processes. Custom scripts (https://github.com/barmadaslab/nuclear-fractionation; *Miguez, 2018c*; copy archived at https://github.com/elifesciences-publications/nuclear-

**Table 1.** Primer sequences used to generate domain deletion mutants.

| Deletion mutation | Amino acids | Primers | Sequences |
| --- | --- | --- | --- |
| ΔZF1 | 288–322 | Forward | 5'-CTT GAA ATC TAC CCA GAA TG-3' |
| | | Reverse | 5'-CTT CGG TAA GAG TCC ATG-3' |
| ΔZF2 | 798–833 | Forward | 5'-CTG AAT AAA TTG GCA GAA GAA C-3' |
| | | Reverse | 5'-AGG TAT CAC ATA GTC TAT ACC-3' |
| ΔRRM1 | 398–473 | Forward | 5'-TAT AAA AGA ATA AAG AAA CCT GAA GG-3' |
| | | Reverse | 5'-GCT AGT TTC CAC TCT GCC-3' |
| ΔRRM2 | 496–575 | Forward | 5'-GTT CTG AGG ATT CCA AAC AG-3' |
| | | Reverse | 5'-TCC AAG CTC TTG CTT TTG-3' |

DOI: https://doi.org/10.7554/eLife.35977.018

**Table 2.** Primer sequences used to generate putative NLS deletion mutants.

| Deletion mutation | Amino acids | Primers | Sequences |
|---|---|---|---|
| ΔpNLS1 | 146–171 | Forward | 5'-AGA GTA CCT AGG GAT GAT TG-3' |
|  |  | Reverse | 5'-AAG CTG TAG AAG GAT TTG G-3' |
| ΔpNLS2 | 473–479 | Forward | 5'-CCT GAA GGA AAG CCA GAT C-3' |
|  |  | Reverse | 5'-CTG GGA TAA ATG AAC TCT CAC-3' |
| ΔpNLS3 | 571–574 | Forward | 5'-CTG GTT CTG AGG ATT CCA ACC-3' |
|  |  | Reverse | 5'-CTC AGA CAG GTC AAC CTT C-3' |
| ΔpNLS4 | 588–611 | Forward | 5'-ACT GAT GGT TCC CAG AAG-3' |
|  |  | Reverse | 5'-CAG TAA ATC AAT GCC TCT G-3' |
| ΔpNLS5 | 701–720 | Forward | 5'-GAG GAA CTT GAT CAA GAA AAC-3' |
|  |  | Reverse | 5'-CAC AGC TTT ATC TGA TGG TTC-3' |
| ΔpNLS6 | 780–784 | Forward | 5'-CAG CCC AAT GTT CCT GTT G-3' |
|  |  | Reverse | 5'-ATA CTC ATC TGG GAT TGT ATA G-3' |
| ΔpNLS7 | 798–833 | Forward | 5'-GAA ACT ATG ACT AGT ACT AGA G-3' |
|  |  | Reverse | 5'-CTG ATA ATG AGG AAG GCT G-3' |
| ΔpNLS4N | 588–595 | Forward | 5'-TCT TAC TCT CCA GAT GGC-3' |
|  |  | Reverse | 5'-CAG TAA ATC AAT GCC TCT G-3' |
| ΔpNLS4C | 608–611 | Forward | 5'-ACT GAT GGT TCC CAG AAG-3' |
|  |  | Reverse | 5'-ATC ACT TGG AGA TTC TTT GC-3' |

DOI: https://doi.org/10.7554/eLife.35977.019

fractionation) were also used to identify and draw bounding ROIs around nuclei of transfected cells based upon MATR3-EGFP or Hoechst 33258 (ThermoFisher, Waltham, MA) fluorescence. Coefficient of variation (CV) was calculated as the standard deviation of fluorescence intensity divided by the mean fluorescence intensity within an ROI.

## Immunocytochemistry

Neurons were fixed with 4% paraformaldehyde, rinsed with phosphate buffered saline (PBS), and permeabilized with 0.1% Triton X-100 in PBS. After brief treatment with 10 mM glycine in PBS, cells were placed in blocking solution (0.1% Triton X-100, 2% fetal calf serum, and 3% bovine serum

**Table 3.** Primer sequences used to generate point mutations.

| Mutation | Primers | Sequences |
|---|---|---|
| Q66K | Forward | 5'-TTC TTC ATT GAA TAA ACA AGG AGC TC-3' |
|  | Reverse | 5'-GAG CTC CTT GTT TAT TCA ATG AAG AA-3' |
| A72T | Forward | 5'-AAG GAG CTC ATA GTA CAC TGT CT-3' |
|  | Reverse | 5'-AGA CAG TGT ACT ATG AGC TCC TT-3' |
| S85C | Forward | 5'-AAT TTG CAG TGT ATA TTT AAC ATT GG-3' |
|  | Reverse | 5'-ATG GGA AGA AGT ACT AGC AGA-3' |
| F115C | Forward | 5'-ATT TTG GCC AGC TGT GGT CTG TCT GCT-3' |
|  | Reverse | 5'-GTT ACT GGC CTG GTC TGC ATC-3' |
| P154S | Forward | 5'-GAA GAA GGC TCT ACC TTG AGT TAT GG-3' |
|  | Reverse | 5'-AGT TCT CCT CCT TTT AAG CTG-3' |
| T622A | Forward | 5'-GAG AGT TCA GCC GAA GGT AAA GAA C-3' |
|  | Reverse | 5'-AGT CTT CTG GGA ACC ATC AGT-3' |

DOI: https://doi.org/10.7554/eLife.35977.020

albumin (BSA), all in PBS) at RT for 1 hr before incubation overnight at 4°C in primary antibody at the following dilutions in blocking solution: rabbit anti-MATR3 (Abcam EPR10635(B); RRID: AB_2491618 for *Figure 1* and EPR10634(B) for *Figure 2*, Cambridge, UK) diluted 1:1000, mouse anti-fibrillarin (Abcam 38F3; RRID: AB_304523) diluted 1:1000, mouse anti-SC35 (Novus Biologicals NB100-1774SS; RRID: AB_526734, Littleton, CO) diluted 1:2000, mouse anti-PML (Santa Cruz Biotechnology sc-377390, Santa Cruz, CA) diluted 1:50. Cells were then washed 3x in PBS, and incubated at RT with secondary antibody, goat anti-rabbit 647 (Jackson ImmunoResearch 111-605-003; RRID: AB_2338072, West Grove, PA) or goat anti-mouse 647 (Jackson ImmunoResearch 115-605-003; RRID: AB_2338902) diluted 1:1000 in blocking solution, for 1 hr at RT. Following 3x rinses in PBS containing 1:5000 Hoechst 33258 dye (ThermoFisher), neurons were imaged by fluorescence microscopy, as described above.

## Fluorescence recovery after photobleaching

Primary neurons were dissected as above and plated in LAB-TEK 8-well borosilicate chambers (ThermoFisher). On DIV 3, they were transfected as before but using 200 µg of pGW1-mApple and 200 µg of pGW1-MATR3-EGFP variants per well. Cell were imaged 2–4 days after transfection using a Nikon A1 confocal microscope operated by Nikon Elements, a 60X objective lens, and a heating chamber with $CO_2$ pre-warmed to 37°C. For MATR3(ΔRRM1/2)-EGFP variants, an ROI corresponding to half of the granule was outlined with Elements and photobleached using a 488 nm laser set at 30% power, 1 pulse/s x 7 s. Fluorescence recovery was monitored up to 10 min after photobleaching. For full-length MATR3 variants, ROIs for photobleaching were drawn in the center of the nucleus for each cell, and recovery was monitored for 6 min.

Image analysis was conducted in FIJI. Rigid body stack registration was used to fix the granules in place relative to the frame. The GFP integrated density for the whole granule was calculated from pre-bleach measurements, as was the fraction of granule integrated density corresponding to the ROI to be photobleached. The decline in this fraction immediately after photobleaching was then calculated and used as the floor, and the return was plotted as the percent recovery within the ROI as a fraction of the original pre-bleach granule integrated density.

Recovery data were fit to the equation $y(t)=A(1-e^{-\tau t})$, where A is the return curve plateau, $\tau$ is the time constant, and t is the time post-bleach. The fitted $\tau$ from each curve was then used to calculate the time to half-return ($t_{1/2}$) using the equation $t_{1/2} = \ln(0.5)/-\tau$. To estimate the diffusion coefficient (D) of these molecules, we used the equation $D = (0.88\ w^2)/(4\ t_{1/2})$, where w is the ROI radius (*Gopal et al., 2017*). This equation assumes spot bleach with a circular stimulation ROI and diffusion limited to the x-y plane. Since we could not be confident that these assumptions were met, we estimated D and downstream parameters by dividing ROI areas by $\pi$ to approximate $w^2$ and solving for D. This estimated value was used in the Einstein-Stokes equation, $D = k_BT/(6\pi\eta r)$, where $k_B$ is the Boltzmann constant, T is temperature in K, $\eta$ is viscosity, and r is the Stokes radius of the particle. As there is no applicable structural data on MATR3, we estimated a Stokes radius of 3.13 nm by applying the MATR3(ΔRRM1/2)-EGFP fusion protein's combined molecular weight of 106.4 kDa to the equation $R_{min} = 0.66M^{1/3}$, where $R_{min}$ is the minimal radius in nm of a sphere that could bound a globular protein with a molecular weight of M (*Erickson, 2009*). Using these constants and the estimated D for each granule, the Einstein-Stokes equation was rearranged to solve for $\eta$.

Photobleaching data from full-length MATR3-EGFP was analyzed in a similar fashion. After calculating the nuclear integrated density, the fraction attributable to photobleaching within the ROI was used for normalization. Intensity data were fit to the $y(t)=A(1-e^{-\tau t})$ equation, $t_{1/2}$ values were calculated as before, and D determined by the equation $D = (0.88\ w^2)/(4\ t_{1/2})$.

## Nuclear/cytoplasmic fractionation and differential solubility

HEK293T cells (STR profiling-validated and mycoplasma-negative; ATCC CRL-3216; RRID: CVCL_0063) were transfected in a 6-well plate with 3 µg of DNA per well using Lipofectamine 2000 according to the manufacturer's instructions. For nuclear/cytoplasmic fractionation, cells were washed with cold PBS 24 hr after transfection, collected with resuspension buffer (10 mM Tris, 10 mM NaCl, 3 mM $MgCl_2$, pH 7.4), and transferred to a pre-chilled 1.5 mL conical tube to sit on ice for 5 min. An equal volume of resuspension buffer with 0.6% Igepal (Sigma, St. Louis, MO) was then added to rupture cell membranes and release cytoplasmic contents, with occasional inversion for 5 min on ice.

Nuclei were pelleted at 100 x g at 4°C for 10 min using a tabletop centrifuge. The supernatant (cytosolic fraction) was collected, and the nuclei were rinsed twice in resuspension buffer without Igepal. To collect nuclear fractions, pelleted nuclei were lysed in RIPA buffer (Pierce) with protease inhibitors (Roche, Mannheim, Germany) on ice for 30 min with occasional inversion. Samples were centrifuged at 9400 x g at 4°C for 10 min, and the supernatant was saved as the nuclear fraction.

For differential solubility experiments, transfected HEK293T were collected in cold PBS 24 hr after transfection and transferred to a pre-chilled conical tube on ice. Cells were then centrifuged at 100 x g for 5 min at 4°C to pellet cells, the PBS was aspirated, and cells were resuspended in RIPA buffer with protease inhibitors. Following lysis on ice for 15 min with occasional inversion, cells were sonicated at 80% amplitude with 5 s on/5 s off for 2 min using a Fisherbrand Model 505 Sonic Dismembrenator (ThermoFisher). Samples were centrifuged at 41,415 x g for 15 min at 4°C to pellet RIPA-insoluble material, with the supernatant removed and saved as the RIPA-soluble fraction. The RIPA-insoluble pellet was washed in RIPA once, and contents resuspended vigorously in urea buffer (7 M urea, 2 M thiourea, 4% CHAPS, 30 mM Tris, pH 8.5). Samples were again centrifuged at 41,415 x g for 15 min at 4°C, and the supernatant was saved as the RIPA-insoluble, urea-soluble fraction.

For SDS-PAGE, stock sample buffer (10% SDS, 20% glycerol, 0.0025% bromophenol blue, 100 mM EDTA, 1 M DTT, 20 mM Tris, pH 8) was diluted 1:10 in lysates and all samples except urea fractions were boiled for 10 min before 5–15 µg of protein were loaded onto 4–15% gradient gels (Bio-Rad, Hercules, CA). For urea fractions, total protein concentration was too low to quantify and so equal volumes of sample across conditions were mixed 1:1 with water and loaded. After electrophoresis, samples were transferred at 30 V overnight at 4°C onto an activated 2 µm nitrocellulose membrane (Bio-Rad), blocked with 3% BSA in 0.2% Tween-20 in Tris-buffered saline (TBST), and blotted overnight at 4°C with the following primary antibodies: rabbit anti-MATR3 (Abcam EPR10634(B)), mouse anti-GAPDH (Millipore Sigma MAB374; RRID: AB_2107445), and rabbit anti-H2B (Novus Biologicals NB100-56347; RRID: AB_838347), all diluted 1:1000 in 3% BSA, 0.2% TBST. The following day, blots were washed in 0.2% TBST, incubated at RT for 1 hr with AlexaFluor goat anti-mouse 594 (ThermoFisher A-11005; RRID: AB_141372) and goat anti-rabbit 488 (ThermoFisher A-11008; RRID: AB_143165), both diluted 1:10,000 in 3% BSA in 0.2% TBST. Following treatment with secondary antibody, blots were washed in 0.2% TBST, placed in Tris-buffered saline, and imaged using an Odyssey CLx Imaging System (LI-COR, Lincoln, NE).

## Statistical analysis

Statistical analyses were performed in R or Prism 7 (GraphPad). For primary neuron survival analysis, the publically available R survival package was used to determine hazard ratios describing the relative survival among populations through Cox proportional hazards analysis. For half-life calculations, mean photoconverted Dendra2 signal in the TRITC channel was measured on a per-cell basis using a custom script (https://github.com/barmadaslab/measurements; *Miguez, 2018b*; copy archived at https://github.com/elifesciences-publications/measurements), log-transformed and fit to linear equation. Photobleaching recovery data were fit to the $y(t)=A(1-e^{-\tau t})$ equation using non-linear regression in R. siRNA knockdown data were plotted using Prism 7, and significance determined via the two-tailed t-test. One-way ANOVA with Tukey's post-test was used to assess for significant differences among nuclear/cytoplasmic ratios, viscosities, D values, and half-lives. Data are shown as mean ± SEM unless otherwise stated.

## Acknowledgements

We thank Dr. Stephen Lentz for his assistance with confocal microscopy, Drs. Claudia Figueroa-Romero and Hilary Archbold for their experimental advice, and Brittany Flores for technical assistance in assembling the manuscript. The present work was supported in part by funding from the National Institutes of Health (NIH) National Institute for Neurological Disorders and Stroke (NINDS) R01 NS097542 (SJB), National Institute for Aging (NIA) P30 AG053760 (SJB), and National Institute of General Medical Sciences (NIGMS) T32 GM007863 (AMM); the Protein Folding Diseases Initiative at the University of Michigan (SJB); the Program for Neurology Research and Discovery (ELF); and the A Alfred Taubman Medical Research Institute (SJB, ELF). Confocal microscopy was performed at the Microscopy and Image Analysis Core of the Michigan Diabetes Research Center funded by NIH

grant P60DK020572 from the National Institute of Diabetes and Digestive and Kidney Diseases (NIDDK).

## Additional information

### Funding

| Funder | Grant reference number | Author |
|---|---|---|
| National Institute of Neurolo-gical Disorders and Stroke | NS097542 | Sami Barmada |
| University of Michigan | | Sami Barmada |
| National Institute on Aging | AG053760 | Sami Barmada |
| National Institute of General Medical Sciences | GM007863 | Ahmed Malik |
| A. Alfred Taubman Medical Research Institute | | Eva Feldman Sami Barmada |
| Program for Neurology Re-search and Discovery | | Eva L Feldman |

The funders had no role in study design, data collection and interpretation, or the decision to submit the work for publication.

### Author contributions

Ahmed M Malik, Conceptualization, Data curation, Formal analysis, Investigation, Visualization, Methodology, Writing—original draft, Writing—review and editing; Roberto A Miguez, Data cura-tion, Software, Methodology; Xingli Li, Investigation, Methodology; Ye-Shih Ho, Conceptualization, Investigation, Methodology; Eva L Feldman, Conceptualization, Funding acquisition; Sami J Bar-mada, Conceptualization, Supervision, Funding acquisition, Visualization, Methodology, Writing—original draft, Project administration, Writing—review and editing

### Author ORCIDs

Ahmed M Malik http://orcid.org/0000-0002-2625-7483
Sami J Barmada http://orcid.org/0000-0002-9604-968X

### Ethics

Animal experimentation: All vertebrate animal work was approved by the Committee on the Use and Care of Animals at the University of Michigan (protocol # PRO00007096). All experiments were carefully planned so that we use as few animals as possible. Pregnant female wild-type, non-trans-genic Long Evans rats (Rattus norvegicus) were housed singly in chambers equipped with environ-mental enrichment. They were fed ad libitum a full diet (30% protein, 13% fat, 57% carbohydrate; full information available at www.labdiet.com), and cared for by the Unit for Laboratory Animal Med-icine (ULAM) at the University of Michigan. Veterinary specialists and technicians in ULAM are trained and approved in the care and long-term maintenance of rodent colonies, in accordance with the NIH-supported Guide for the Care and Use of Laboratory Animals. All rats were kept in routine housing for as little time as possible prior to euthanasia and dissection, minimizing any pain and/or discomfort. Pregnant dams were euthanized by $CO_2$ inhalation at gestation day 20. For each animal, euthanasia was confirmed by bilateral pneumothorax. Euthanasia was fully consistent with the rec-ommendations of the Guidelines on Euthanasia of the American Veterinary Medical Association and the University of Michigan Methods of Euthanasia by Species Guidelines. Following euthanasia, the fetuses were removed in a sterile manner from the uterus and decapitated. Primary cells from these fetuses were dissected and cultured immediately afterwards.

### Decision letter and Author response

Decision letter https://doi.org/10.7554/eLife.35977.023

Author response https://doi.org/10.7554/eLife.35977.024

## Additional files

### Supplementary files
• Transparent reporting form
DOI: https://doi.org/10.7554/eLife.35977.021

### Data availability
Survival and intensity data have been provided for Figures 1, 2, 3, 4, 6 & 7.

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
