## [Decision Letter]

Thank you for submitting your article "Matrin 3-dependent neurotoxicity is modified by nucleic acid binding and nucleocytoplasmic localization" for consideration by *eLife*. Your article has been reviewed by three peer reviewers, including J Paul Taylor as the Reviewing Editor and Reviewer #1, and the evaluation has been overseen by a Senior Editor.

The reviewers have discussed the reviews with one another and the Reviewing Editor has drafted this decision to help you prepare a revised submission

Summary:

This is a thorough and carefully conducted study of Matrin 3 (MATR3), including analysis of a series of disease-causing MATR3 mutations in neurons. Whereas the details of the molecular mechanism of MATR3-mediated neurodegeneration remains obscure, the study generates valuable basic insights into MATR3 biology, including the ability to engage in nuclear liquid bodies, how this is influenced by nucleic acid interaction, and the basis of nuclear localization. The study also sheds some light onto the probable pathogenic mechanism, revealing it to be a dominant effect within the nucleus. The approaches taken in this study are powerful because they permit longitudinal evaluation with rigorous application of statistical analyses that permit assessment of dose-dependent consequences of MATR3 expression, including the impact of disease-associated mutations. Whereas some of these insights were very recently published, this paper has novel elements and it is a service to the field to show such rigorous corroboration of those elements that are replicated.

Essential revisions:

• The presentation of the data indicating that the toxicity associated with altered MATR3 levels is dose-responsive was a little convoluted and left one reviewer uncertain as to whether there was in fact evidence of a dose response. Please clarify in the revision: how were MATR3 levels assessed and how was the correlation between MATR3 expression level and risk of death established?

• Since the **Δ**pNLS4N lies within the RRM2, how can you exclude that this deletion does not affect RNA binding? Why is the **Δ**RRM2 (Figure 3D) not showing increased cytoplasmic localization, like the **Δ**pNLS4N (Figure 5C) does? I find this puzzling, since the deleted amino acids in the **Δ**pNLS4N lie within the RRM2.

• The effect of the S85C mutant on the dynamics of the nuclear droplets formed by deletion of RRM2 is very interesting. However, to generalize this observation, it would be important to show the same effect with another disease-related mutant within the same low complexity region in the N-terminus of the protein. Perhaps examine an additional mutation in this region that does not introduce a novel cysteine.

---

## [Author Response]

Essential revisions:• The presentation of the data indicating that the toxicity associated with altered MATR3 levels is dose-responsive was a little convoluted and left one reviewer uncertain as to whether there was in fact evidence of a dose response. Please clarify in the revision: how were MATR3 levels assessed and how was the correlation between MATR3 expression level and risk of death established?

In an effort to clarify the investigation of MATR3’s dose-dependent effects on neurodegeneration, we added new data and substantially revised this portion of the manuscript. First, we performed experiments to better demonstrate the means by which we estimate MATR3 abundance in transfected neurons. Transient transfection delivers a different amount of vector to each cell, resulting in substantial variability in protein expression for individual cells. Such variability can be difficult to appreciate using population-based approaches such as Western blotting but are readily visualized by single-cell techniques including immunofluorescence (Arrasate et al., 2005; Miller et al., 2010; Barmada et al., 2015). Therefore, to estimate the degree of MATR3 overexpression in individual neurons, we measured MATR3 antibody reactivity by quantitative immunofluorescence in neurons transfected with EGFP or MATR3-EGFP (Figure 1D). We found that MATR3-EGFP transfected cells had approximately 2.8-fold higher MATR3 immunoreactivity than untransfected cells and those transfected with EGFP alone (Figure 1E). Further, in agreement with previous work relating single-cell fluorescence intensity to immunoreactivity (Arrasate et al., 2004), we detected a linear relationship between EGFP fluorescence intensity and anti- MATR3 antibody reactivity in individual neurons expressing MATR3-EGFP (Figure 1F). These data confirm that GFP intensity provides a reliable, single-cell estimate of EGFP or MATR3-EGFP expression.

We also took advantage of this relationship to analyze the consequences of EGFP or MATR3-EGFP expression (measured 24 h after transfection) on neuronal survival using penalized splines (Miller et al., 2010; Barmada et al., 2015). These models enable us to predict the impact of single-cell protein expression on the risk of death within separate populations of cells expressing either EGFP (Figure 1G) or MATR3-EGFP (Figure 1H). Consistent with the results of prior studies, we detected a reduced risk of death in association with higher EGFP expression levels (Miller et al., 2010), implying that unhealthy or dying neurons are unable to express high amounts of EGFP. Conversely, we noted a significant increase in the risk of death for cells expressing high levels of MATR3(WT)-EGFP (Figure 1H); this relationship is similar to that observed for other proteins associated with neurodegenerative disorders, including TDP-43 (ALS/FTD; Barmada et al., 2015) and mutant huntingtin (Huntington’s disease; Miller et al., 2010). Taken together, these data support a dose- dependent toxicity of MATR3-EGFP in primary neurons.

*• Since the*
***Δ****pNLS4N lies within the RRM2, how can you exclude that this deletion does not affect RNA binding? Why is the*
***Δ****RRM2 (Figure 3D) not showing increased cytoplasmic localization, like the*
***Δ****pNLS4N (Figure 5C) does? I find this puzzling, since the deleted amino acids in the*
***Δ****pNLS4N lie within the RRM2.*

As depicted in Figure 6A, the NLS4N sequence (designated “pNLS4N” in the revised manuscript) begins downstream of RRM2. Therefore, we would not expect deletion of this sequence to interfere with RNA binding; conversely, RRM2 deletion would not be expected to perturb nuclear localization. Rather, it is NLS3 (now “pNLS3”) that lies inside RRM2 and whose deletion results in the formation of small intranuclear droplets.

These observations are consistent with the redistribution of MATR3(∆RRM2)-EGFP into similar intranuclear droplets (Figure 4). However, we did not detect any change in MATR3 nucleocytoplasmic localization upon deletion of pNLS3, suggesting that it does little to influence MATR3 nuclear import.

• The effect of the S85C mutant on the dynamics of the nuclear droplets formed by deletion of RRM2 is very interesting. However, to generalize this observation, it would be important to show the same effect with another disease-related mutant within the same low complexity region in the N-terminus of the protein. Perhaps examine an additional mutation in this region that does not introduce a novel cysteine.

The reviewer raises an interesting question regarding the influence of the MATR3 N-terminus on the behavior of nuclear droplets formed by RNA binding-deficient MATR3 variants. To answer this question, we cloned Q66K and A72T—two pathogenic MATR3 mutations responsible for familial ALS (Lin et al., 2015; Maranagi et al., 2017) and located within the protein’s N-terminus—on the MATR3(∆RRM1/2)-EGFP background. As before, we imaged transfected neurons by confocal fluorescence microscopy, bleached the resulting intranuclear MATR3 granules, and measured the recovery of fluorescence intensity over time to estimate MATR3 mobility (Figure 5—figure supplement 1). Although we noted a trend towards enhanced viscosity associated with each of these mutations, the effect did not reach significance.

We also expand our interpretation of these findings within the Discussion section and comment on the recent report by another group suggesting that the S85C MATR3(∆RRM2) does not form granules in C2C12 myoblasts (Iradi et al., 2018):

“In C2C12 myoblast cells, MATR3 forms intranuclear spherical structures with liquid-like properties upon deletion of RRM2, though these granules were smaller and more numerous than those we detected in primary neurons, a difference that may be due to expression level and cell type (Iradi et al., 2018). […] In the absence of effective RNA binding, therefore, MATR3 may be free to interact with other proteins and itself through its low-complexity domains, driving LLPS.”